# Simplifying Latent Dynamics with Softly State-Invariant World Models

**Tankred Saanum**[1†]     **Peter Dayan**[1,2]     **Eric Schulz**[1,3]

[1]Max Planck Institute for Biological Cybernetics,   [2]University of Tübingen
[3] Helmholtz Institute for Human-Centered AI, Helmholtz Center Munich, Neuherberg, Germany
[†]`tankred.saanum@tuebingen.mpg.de`

## Abstract

To solve control problems via model-based reasoning or planning, an agent needs to know how its actions affect the state of the world. The actions an agent has at its disposal often change the state of the environment in systematic ways. However, existing techniques for world modelling do not guarantee that the effect of actions are represented in such systematic ways. We introduce the Parsimonious Latent Space Model (PLSM), a world model that regularizes the latent dynamics to make the effect of the agent's actions more predictable. Our approach minimizes the mutual information between latent states and the *change* that an action produces in the agent's latent state, in turn minimizing the dependence the state has on the dynamics. This makes the world model softly state-invariant. We combine PLSM with different model classes used for *i*) future latent state prediction, *ii*) planning, and *iii*) model-free reinforcement learning. We find that our regularization improves accuracy, generalization, and performance in downstream tasks, highlighting the importance of systematic treatment of actions in world models.

## 1   Introduction

In Reinforcement Learning (RL), the actions that an agent can use to solve tasks often have systematic and predictable effects on the state of the environment. When stepping on the gas pedal, the car tends to accelerate, when holding down the joystick in a given direction, the video game character tends to move in that direction, and so forth. These typical effects have exceptions that are predictable as well, e.g. stepping on the gas will not lead to acceleration if the car engine is turned off, and the video game character will not move if it is facing a wall. How can we learn world models that capture these systematic properties of actions? World models predict the agent's future states, given the current state and action [1, 2, 3]. Most approaches to world modelling represent high-dimensional observations (such as images) in compact, low-dimensional latent states $\mathbf{z}_t$, simplifying model-based prediction and control [4]. Here we explore the possibility of compressing states and dynamics jointly to learn systematic effects of actions (see Fig. 1). As is common practice in many dynamics model architectures [4, 5], we consider the case where the model predicts the next latent $\tilde{\mathbf{z}}_{t+1}$ state by predicting the *difference* $\tilde{\Delta}_t^{\mathbf{a}_t}$, or the *change*, between the current and future latent state, given an action $\mathbf{a}_t$.

$$\tilde{\mathbf{z}}_{t+1} = \mathbf{z}_t + \tilde{\Delta}_t^{\mathbf{a}_t} \tag{1}$$

Even if $\mathbf{z}_t$ is low-dimensional, the effects of actions might not be represented parsimoniously within the world model: Performing the *same* action $\mathbf{a}_t$ in two similar states $\mathbf{z}, \mathbf{z}'$ might produce two very different deltas $\Delta, \Delta'$, due to how the observation encoder has constructed the latent state space (see

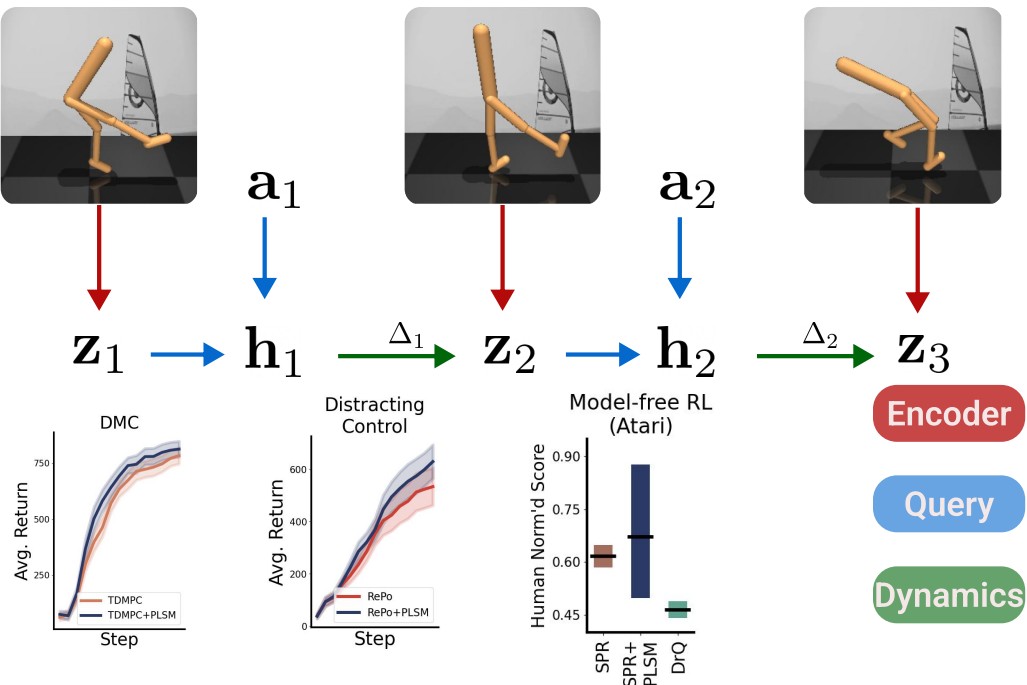

Figure 1: **Overview**: World models are commonly used to predict latent trajectories, predict sequences of pixel observations, and perform planning. We propose an architecture together with an information bottleneck for learning simple and parsimonious world models. Our method relies on a query network that extracts a sparse representation $\mathbf{h}_t$ for predicting latent transition dynamics. Combining our method with auxiliary loss functions for *i*) contrastive learning *ii*) planning and *iii*) and model-free RL, we see consistent performance improvement in all domains. Lines and bars show mean performance from three sets of RL benchmarks. Error bars represent 95% confidence interval.

Fig. 2). We introduce the Parsimonious Latent Space Model, or PLSM for short, a world model where actions have more predictable effects on the inferred latent states of the agent. Dynamics are simplified by minimizing how much the predicted dynamics $\tilde{\Delta}_t^{\mathbf{a}_t}$ depend on $\mathbf{z}_t$. This pushes the world model to represent states in a way that makes actions have coherent and predictable effects on the dynamics. We still allow the dynamics to vary depending on the state, but we penalize the extent of this dependence, resulting in dynamics that are softly state-invariant.

We combine PLSM with two classes of world models: Contrastive World Models (CWM) [3] for latent state prediction, and with Self Predictive Representations (SPR) for model-free and model-based control (TD-MPC) [6, 5, 7]. Across control experiments and prediction experiments we see improvements in planning, representation learning for control, robustness to noise, world model accuracy, and generalization.

## 2 Latent dynamics

We assume that sequences of states, actions and rewards arise in a Markov Decision Process (MDP). An MDP consists of a state space $\mathcal{S}$, an action space $\mathcal{A}$, and transition dynamics $\mathbf{s}_{t+1} \sim P(\mathbf{s}_{t+1}|\mathbf{s}_t, \mathbf{a}_t)$ determining how the state evolves with the actions the agent performs. In RL, we additionally care about the reward function $r(\mathbf{s}_t, \mathbf{a}_t)$, which maps state-action pairs to a scalar reward term. Here, the goal is to learn the policy $\pi_\theta(\mathbf{a}_t|\mathbf{s}_t)$ that maps states to the actions with the highest possible $Q$-values $Q_{\pi_\theta}(\mathbf{s}_t, \mathbf{a}_t) = \mathbb{E}_{\pi_\theta}\left[\sum_{t=1}^T \gamma^t r(\mathbf{s}_t, \mathbf{a}_t)\right]$, where $\gamma$ is a discount factor. In this paper, we consider latent dynamics learning both in reward-free and RL settings.

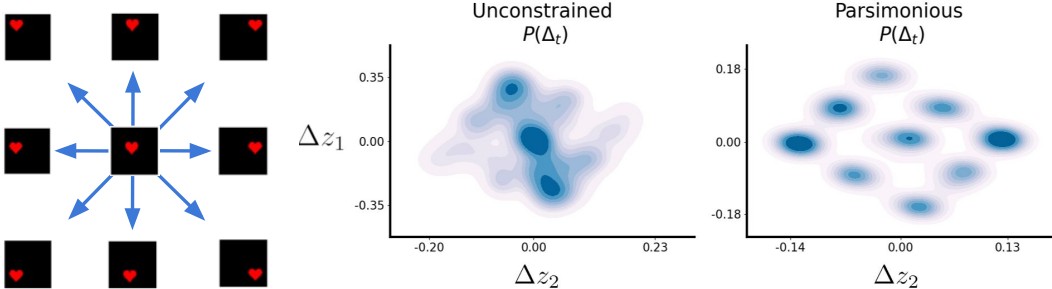

Figure 2: The heart (left) can appear on any $x, y$ coordinate in a two-dimensional latent space with boundaries, on which it can transition in 9 different ways (moving in eight directions and standing still, for instance when moving into a boundary). Encouraging dynamics to be parsimonious recovers these 9 different possible transitions (see right), whereas an unconstrained model (see center) does not.

To predict environment dynamics, the current state $\mathbf{s}_t$ is first transformed using an encoder into a latent state $\mathbf{z}_t$ that compactly represents the agent's sensory observation (2). The world model predicts the *change* in the latent state, $\tilde{\Delta}_t^{\mathbf{a}_t}$ that is induced by the action $\mathbf{a}_t$ (3) & (4).

$$\mathbf{z}_t = e_\theta(\mathbf{s}_t) \tag{2}$$

$$\tilde{\Delta}_t = d_\theta(\mathbf{z}_t, \mathbf{a}_t) \tag{3}$$

$$\tilde{\mathbf{z}}_{t+1} = \mathbf{z}_t + \tilde{\Delta}_t \tag{4}$$

We omit the action superscript from $\tilde{\Delta}_t$ for simplicity. Here, $e_\theta$ is the encoder network mapping states to latent states, and $d_\theta$ is the dynamics network mapping $\mathbf{z}_t$ and $\mathbf{a}_t$ to $\tilde{\Delta}_t$.

## 2.1 Parsimonious latent dynamics

Consider the probability distribution of transitions $P(\tilde{\Delta}_t|\mathbf{a}_t)$ given the agent's actions, marginalizing across latent states. If actions have predictable effects on the state of the environment, the entropy of $P(\tilde{\Delta}_t|\mathbf{a}_t)$ will be low – knowing the latent state gives little information about the effect of $\mathbf{a}_t$. To make the world model handle actions more systematically, we propose to minimize the amount of information the world model needs from $\mathbf{z}_t$ in order to predict correctly how an action changes the state of the world. This quantity is represented in the mutual information between the latent state $\mathbf{z}_t$ and the dynamics $\tilde{\Delta}_t$:

$$I(\mathbf{z}_t; \tilde{\Delta}_t|\mathbf{a}_t) = \mathcal{H}[\tilde{\Delta}_t|\mathbf{a}_t] - \mathcal{H}[\tilde{\Delta}_t|\mathbf{z}_t, \mathbf{a}_t] \tag{5}$$

where $\mathcal{H}[\cdot]$ denotes the Shannon entropy. If this quantity is 0, the latent dynamics $\tilde{\Delta}_t$ only depend on the agent's *action* $\mathbf{a}_t$ and not $\mathbf{z}_t$. In this extreme, all $\tilde{\Delta}_t$ are predicted exclusively by the action. However, it is rarely the case that the action can capture an environment's full dynamics, and making dynamics contingent on states is often necessary to some degree.

To allow only the relevant information from $\mathbf{z}_t$ to influence the dynamics, we introduce a query network $f_\theta$ which maps latent state-action pairs to a latent code $\mathbf{h}_t$. We modify the next-step prediction components accordingly

$$\mathbf{h}_t = f_\theta(\mathbf{z}_t, \mathbf{a}_t) \tag{6}$$

$$\tilde{\Delta}_t = d_\theta(\mathbf{h}_t, \mathbf{a}_t) \tag{7}$$

We give the query network information about the action that the transition is conditioned on. This allows the network to attend to the relevant bits in $\mathbf{z}_t$ to output an appropriate $\mathbf{h}_t$. Finally, to make

$\mathbf{h}_t$ represent only the *minimal* amount of information needed to predict the next state, provided that $\mathbf{a}_t$ is known, we penalize the norm of $\mathbf{h}_t$ [8, 9]. This type of regularization has been used to constrain representations of deterministic Autoencoders in past work [8, 9], with [8] showing that it is equivalent to minimizing the KL divergence to a constant variance zero mean Gaussian. Other regularizers and stochastic formulations are also possible (see Appendix C). For simplicity we use the deterministic variant and leave stochastic versions for future work.

The strength of the penalization is controlled through a hyperparameter $\beta$. This regularization minimizes how much $\mathbf{h}_t$ can vary with $\mathbf{z}_t$ and hence their mutual information (see Appendix A). We then train our encoder $e_\theta$, query network $f_\theta$, and dynamics $d_\theta$ jointly to minimize the following information regularized loss function.

$$\mathcal{L} = ||e_\theta(\mathbf{s}_{t+1}) - (\mathbf{z}_t + d_\theta(\mathbf{h}_t, \mathbf{a}_t))||_2^2 + \beta||\mathbf{h}_t||_2^2 \tag{8}$$

Our loss function encourages that the mutual information term is kept as low as possible, while still allowing the model to predict the next latent state accurately. In contrast to information bottlenecks imposed on the latent states themselves, we apply an information bottleneck to the dynamics, making $\tilde{\Delta}_t$ easier to predict simply given $\mathbf{a}_t$. Regularizing $\mathbf{h}_t$ differs from regularizing $\mathbf{z}_t$ in important ways. In environments where the dynamics $\Delta$ can be predicted perfectly from the actions and independently of the state, regularizing $\mathbf{h}_t$ will not lead to a loss in information in the latent representation $\mathbf{z}_t$. This is because the bottleneck on $\mathbf{h}_t$ only constrains the model in using information from $\mathbf{z}_t$ to predict $\Delta_t$, and not necessarily in predicting $\mathbf{z}_{t+1}$. See Appendix B for a comparison of our method against $L_1$ and $L_2$ norm regularization on latent states. Notably, our method can *also* lead to state compression, in that $e_\theta$ will be encouraged to omit features from $\mathbf{s}_{t+1}$ that cannot be predicted easily with little information from $\mathbf{z}_t$.

Unfortunately, the above loss function has a trivial solution: it can be minimized completely if $d_\theta$ and $e_\theta$ output a constant $\mathbf{0}$ vector for all states and state-action pairs [7]. This issue is referred to as *representational collapse*. Representational collapse can be remedied in various ways. To show the generality of our information bottleneck, we combine it with two different approaches for mitigating representational collapse, a self-supervised approach for model-based and model-free RL in Section 3, and a contrastive approach for future state prediction in Section 4.

## 3 Parsimonious dynamics for Reinforcement Learning

### 3.1 Model-based RL

We evaluated the PLSM's effect on planning algorithms' ability to learn policies in continuous control tasks. To do so, we built upon the TD-MPC algorithm [6], an algorithm that jointly learns a latent dynamics model and performs policy search by planning in the model's latent space.

TD-MPC makes use of a Task-Oriented Latent Dynamics (TOLD) model. This dynamics model is trained to predict its own future state representations from an initial state and action sequence while making sure that the controller's policy $\pi_\theta$ and $Q$-value function are decodable from the latent state (hence the name *task-oriented* latent dynamics). TOLD falls under the category of Self Predictive Representation (SPR) models, since it uses an exponentially moving target encoder $e_\theta^-$ with the stop-gradient operator to learn to predict its own representations. For planning TD-MPC uses the Cross-Entropy Method [10], searching for actions that maximize $Q$-values.

Only minimal adjustments to the TOLD model are necessary to attain parsimonious dynamics. Instead of predicting the next latent directly from the current latent and action $\tilde{\mathbf{z}}_{t+1} = d_\theta(\mathbf{z}_t, \mathbf{a}_t)$, we use a query network $f_\theta$, mapping latent state-action tuples to $\mathbf{h}_t$ and then minimize

$$\mathcal{L}_{\text{SPR}} = ||\text{sg}(e_\theta(\mathbf{s}_{t+1})) - (\mathbf{z}_t + d_\theta(\mathbf{h}_t, \mathbf{a}_t))||_2^2 + \beta||\mathbf{h}_t||_2^2$$

We evaluated the efficacy of parsimonious dynamics for control in five state-based continuous control tasks from the DeepMind Control Suite (DMC) [11]. We chose the following environments: *i*) acrobot-swingup, due to its challenging and chaotic dynamics. *ii*) finger-turn hard, which poses a challenging exploration problem that TD-MPC was found to struggle with. *iii*) quadruped-walk, *iv*) quadruped-run and *v*) humanoid-walk due to the high-dimensional dynam-

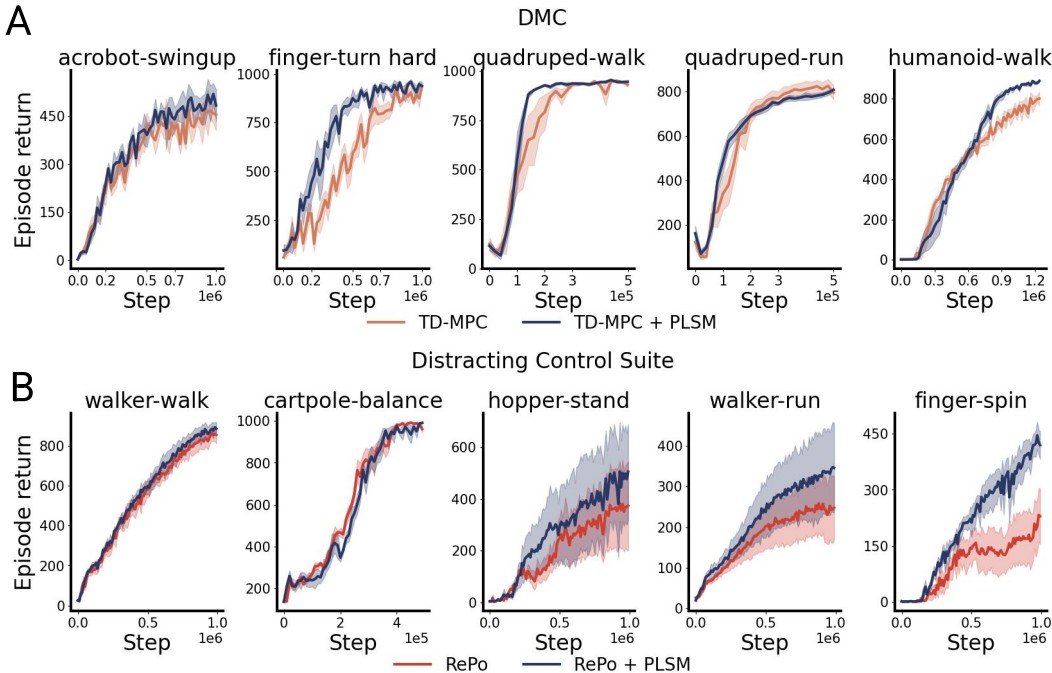

Figure 3: PLSM, when incorporated into either the TD-MPC algorithm (**A**), or RePo (**B**), improves planning in continuous control tasks with high-dimensional and complex dynamics, and visual distractions, respectively. Lines show the average return attained across 15 evaluation episodes, averaged over five seeds. The shaded region represents the 95% confidence interval.

ics. These tasks have dynamics that appear complex in the original state-space but could potentially be simplified in an appropriate latent space by introducing a dynamics bottleneck.

We trained the latent dynamics and planning models in the five tasks up to a million environment steps. Scores for TD-MPC are obtained from the original implementation provided by the authors[1]. Again we used $\beta = 0.1$ for all tasks except for `humanoid-walk`, where $\beta = 0.001$ was more successful. Otherwise we relied on the standard hyperparameters from [6]. We see clear performance gains due in all tasks except `quadruped-run` (Fig. 3A). Our results suggest that modeling the world with simple dynamics can be beneficial for RL and trajectory optimization.

## 3.2 Distracting visual control

Since our regularization compresses away aspects of the environment whose dynamics are unpredictable given the agent's actions, it could be beneficial in control tasks with distractors. We implemented PLSM on top of RePo [12], relying on the authors' official implementation[2]. RePo is a model-based RL algorithm based on the Dreamer [2] architecture. Here the environment dynamics are represented through a GRU network which is updated recurrently with latent state and action variables. To make the dynamics parsimonious, we update the GRU using a compressed query representation $\mathbf{h}_t$ subject to $L_2$ regularization instead of the full state representation $\mathbf{z}_t$, resulting in recurrent dynamics that are softy state-invariant. We then trained RePo with and without our regularization on the Distracting Control Suite [13], a challenging visual control benchmark based on DMC, where the background is replaced with a random, distracting video from the 2017 Davis video dataset. These videos are independent of the agent's actions, irrelevant for rewards, and change from episode to episode.

We trained RePo with and without the PLSM objective in five Distracting Control Suite tasks for one million environment steps across five seeds. We used a regularization coefficient of $\beta = $ 1e-7

---

[1]See https://github.com/nicklashansen/tdmpc
[2]See https://github.com/zchuning/repo

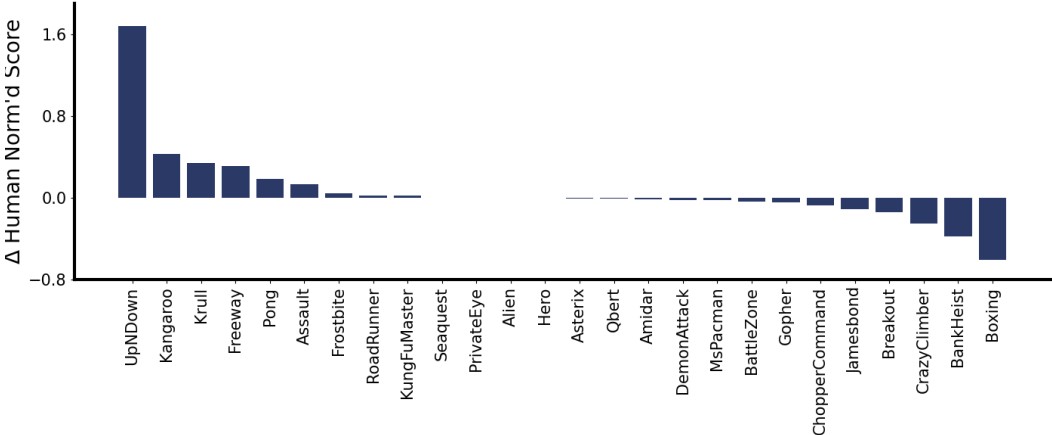

Figure 4: Changing the dynamics model in SPR to PLSM increases score in several Atari games, with little implementation overhead. On average, human normalized scores are higher when using PLSM dynamics. Bars show difference in human normalized score between SPR with and without PLSM dynamics, averaged over five seeds.

for all tasks. This produced a considerable improvement over RePo in more challenging tasks like `hopper-stand`, `walker-run` and `finger-spin` (see Fig. 3B). Our results suggest that encouraging the dynamics model to represent the effects of actions more consistently can improve its generalization ability in environments with distractors.

### 3.3 Model-free RL

Next we tested whether the learned latent space of PLSM could provide useful for model-free learning. Several methods rely on latent dynamics learning as an auxiliary objective for model-free RL [5, 7, 14, 15]. Since PLSM arranges the latent space in way that makes state transitions more predictable, it may discover useful state features and ignore aspects of the environments that would make the dynamics unpredictable otherwise. We build upon the SPR implementation for Atari[3] [16], which uses a latent dynamics model for next latent state prediction. We alter the architecture of this latent dynamics model in the same way we did for TD-MPC, and add the $h_t$ norm to the loss function. Setting $\beta = 5$ and leaving all other hyperparameters as per the standard implementation, we train the PLSM augmented SPR algorithm on 100k environment steps across 5 seeds on all 26 Atari games. We use the SPR scores reported in [17] as our baseline.

Across several games we see substantial improvements to human normalized score (see Fig. 4). Averaging over all games, using PLSM dynamics improves human normalized scores by 5.6 percentage points (61.5 % for SPR vs 67.1% with PLSM). While we see improvements in games such as UpNDown and Kangaroo, there are other games where the regularization impacts performance negatively. Performance could potentially improve by fine-tuning the regularization strength for these domains. See Appendix I for the full score table.

## 4 Future state prediction

We found that our regularization improved model-free and model-based performance across several environments. Next we evaluated whether PLSM also generally improves world models' long-horizon prediction accuracy in latent space. Using the evaluation framework and environments from [3] (see Fig. 13 for example observations), we generated datasets of image, action, next-image triplets from two Atari games (Pong and Space Invaders), and grid worlds with moving 3D cubes and 2D shapes, where each action corresponds to moving an object in one of four cardinal directions. To make the learning tasks more challenging, we increased the number of movable objects from 5 to 9. Additionally we created an environment based on the dSprite dataset [18], with four sprites

---

[3]See https://github.com/mila-iqia/spr

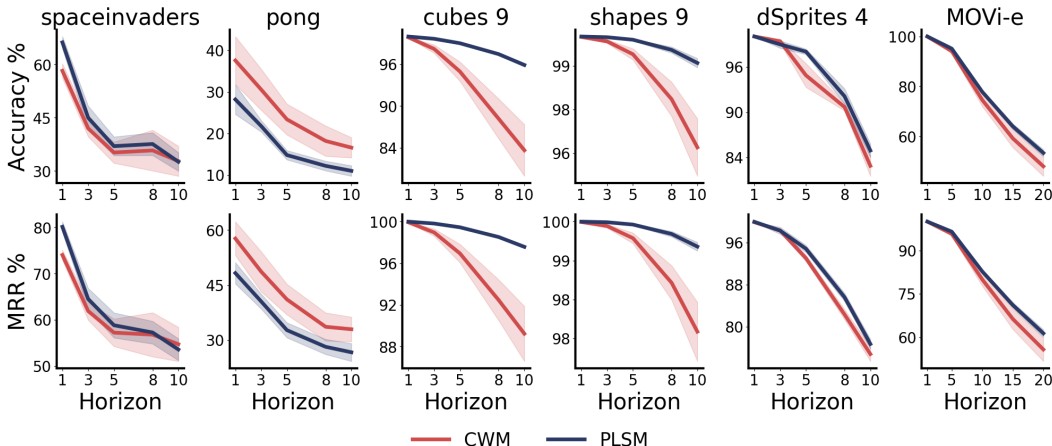

Figure 5: PLSM improves contrastive world models' accuracy in long-horizon latent prediction in five out of six environments. In the cubes and shapes dataset, the PLSM is close to perfect even when predicting as far as 10 timesteps in the future. Lines show accuracy on entire test data averaged over five random seeds. The shaded region corresponds to the standard error of the mean.

traversing latent generative factors on a random walk. The sprites had 6 generative factors: Spatial $x, y$ coordinates, scale, rotation, color, and shape. The sprites could vary in coordinates, scale, and rotation within episodes, and additionally vary in color across episodes. Lastly, we evaluate PLSM on a dynamic object interaction dataset with realistic textures and physics without actions, MOVi-E [19], to see if our method can be beneficial in action-free settings.

Following [3], we pair PLSM with a contrastive loss function to mitigate representational collapse: The contrastive loss encourages that different states are *distinguishable* in the latent space. Given a latent state and action, the model should minimize the distance between the predicted and true future latent state, while maximizing the distance between the predicted future latent state $\mathbf{z}_{t+1}$ and all other latent states in the training batch $\mathbf{z}^-$, up to a margin $\lambda$.

$$\mathcal{L}_{\text{Contrastive}} = ||\mathbf{z}_{t+1} - \tilde{\mathbf{z}}_{t+1}||_2^2 + \max(0, \lambda - ||\mathbf{z}^- - \tilde{\mathbf{z}}_{t+1}||_2^2) \tag{9}$$

To apply our regularization on the contrastive dynamics model, we simply add the norm of the query representation to the contrastive loss, similarly to equation (8). We fitted the regularization coefficient $\beta$ with a grid search and found a value of $0.1$ to work the best. As a baseline we used the unregularized contrastive model from [3], referred to as CWM (for Contrastive World Model), and its dynamics are defined through Equation (2), (4) and (9). We also combine PLSM with the slot-based version of this model, called C-SWM (see Appendix H for results).

The models were scored based on their ability to correctly predict future states (e.g. latent prediction accuracy). The models were trained and evaluated using the same parameters and metrics as in [3]: Given a state $\mathbf{s}_t$, a sequence of $N$ actions $\mathbf{a}_t, ..., \mathbf{a}_{t+N-1}$, and the resulting state $\mathbf{s}_{t+N}$, we make the model predict its latent representation of $\mathbf{s}_{t+N}$ from the initial state and action sequence. The models were evaluated at several prediction horizons. Finally, we report the Hits at Rank 1 accuracy for transitions in the test set, a common test metric for contrastive models [3, 20, 21].

### 4.1 PLSM improves long-horizon prediction accuracy

PLSM could better predict its own representations further into the future in five out of the six datasets (Fig. 5). We see the greatest gains in the cubes and shapes environments. Here, all 9 objects can collide with each other and the grid boundaries depending on their position. Always considering all possible interactions makes it challenging for a model to generalize to novel transitions. A more parsimonious solution is simply to represent whether or not the object in question would collide or not if moved in the direction specified by $\mathbf{a}_t$. Our results suggest that our regularization can help learn such representations, affording better generalization to left-out transitions.

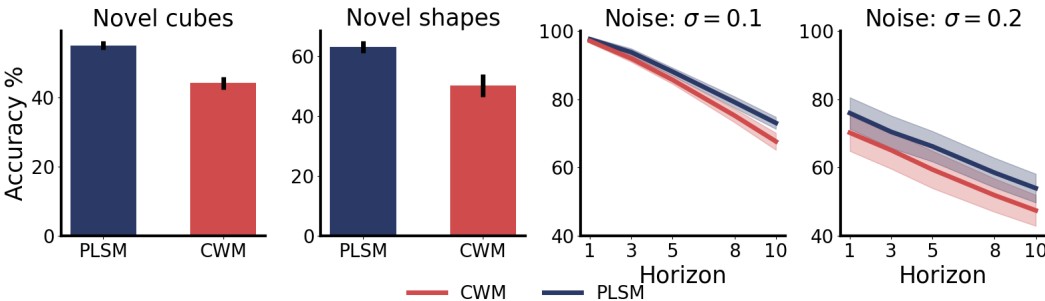

Figure 6: PLSM improves generalization and robustness in contrastive models: When exposed to scenes with fewer objects than trained on (cubes and shapes environment), or corrupted data (dSprite environment) from the test set, PLSM improves accuracy over the CWM. Lines represent the average of models trained across five seeds. Shaded regions and bars reflect the standard error of the mean.

In one environment, Pong, we do not see an advantage in encouraging parsimonious dynamics. Here, various components are outside of the agent's sphere of influence, for instance, the movement of the opponent's paddle. This makes it challenging for the PLSM to capture all aspects of the environment state in its dynamics. In environments with non-controllable dynamics, we offer a remedy by only enforcing half of the latent space to be governed by parsimonious dynamics, and allowing the other half to be unconstrained. This hybrid model in turn shows the strongest performance in the Atari environments. See Appendix G for details.

## 4.2 Generalization and robustness

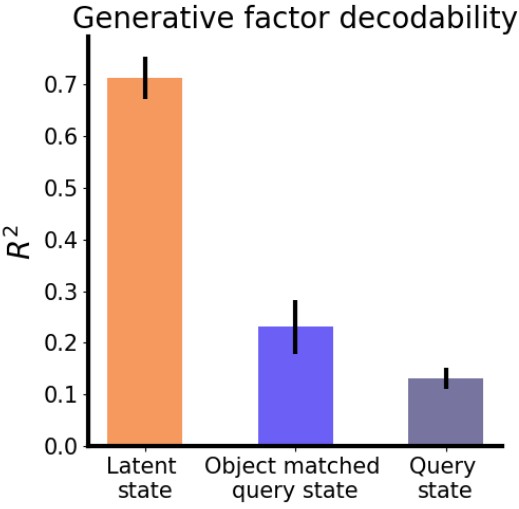

Figure 7: Latent states $\mathbf{z}_t$ carry decodable information about the data generating factors, whereas query states $\mathbf{h}_t$ do not. When conditioning on an action $\mathbf{a}_t$, query states carry more information about the object that the action changes.

Next, we evaluated the generalization and robustness properties of PLSM. For the cubes and shapes environments, we generated novel datasets where the number of moving objects was lower than in the original training data. In these datasets, the number of moving objects varied from 1 to 7. We also probed the models' robustness to noise. We corrupted test data from the dSprite environment with noise sampled from Gaussian distributions. Models were tested both in a low noise ($\sigma = 0.1$) and a high noise ($\sigma = 0.2$) condition. See Supplementary Fig. 14 for example data.

When tested on scenes with fewer objects than the training data, we see a general decrease in accuracy because of the domain shift. Still, with our information-theoretic dynamics bottleneck the model generalizes significantly better to the out-of-distribution transition data (Fig. 6). Since PLSM seeks to predict the next state using as little information from its latent representations as possible, it is more likely to learn that the blocks move in a way that is generally invariant to the number of other blocks in the scene.

Lastly, PLSM proved more robust to Gaussian noise than the unconstrained dynamics models: In the high noise condition, PLSM dynamics still accurately predict almost 60% of the transitions after 10 steps, whereas the unconstrained dynamics models, more susceptible to use noisy information in the state, predict less than 50% accurately.

We also investigated the representations learned by the query network (see Fig. 7). To verify that PLSM learns objects' attributes (such as position and orientation), we attempted to decode ground truth object positions, scales and orientations in the dSprite dataset from PLSM latent states. We find that one can decode these ground truth factors with high accuracy from the latents. However, trying to decode these variables from the query states yielded substantially lower accuracy, indicating that they contain less information about the generative factors of the environment. Interestingly, when we condition PLSM on an action that affects only one object, we can decode attributes of that object more accurately from the query state than average. We observe this because the query state is designed to only encode information that is relevant to predict the effect of individual actions.

In sum, parsimonious dynamics regularization improves both generalization and robustness properties in the three environments tested. The mutual information bottleneck on $\tilde{\Delta}_t$ therefore not only improves prediction accuracy for data in the training distribution, but may also allow the model to generalize better to out-of-distribution data, and improve its robustness to noisy observations.

## 5   Related work

Our approach introduces a mutual information constraint between the latent states $\mathbf{z}_t$ and the latent dynamics $\tilde{\Delta}_t$ inferred by the model. Several methods focus on state compression for dynamics modeling [1, 22, 5, 6, 23, 24]: The Recurrent State Space Model (RSSM) [22, 2, 25], uses a variational Auto-Encoder in combination with a recurrent model (e.g. a GRU [26]) to infer compact latent states in partially observable environments. Latent consistency is enforced by minimizing the Kullback-Leibler (KL) divergence between latent states predicted by the model and latent states inferred from pixels. The KL term regularizes the latent state not to contain more information than can be predicted by the dynamics model [2, 27]. Unlike our approach, this information bottleneck is not applied to the dynamics $\tilde{\Delta}_t$ themselves, but to the representations $\mathbf{z}_t$. Expanding on this line of work, RePo [12] discards image reconstruction from the RSSM, and simply enforces that the model can reconstruct the environment's reward function, leading to stronger compression and improved performance in tasks with unpredictable elements. Similar approaches like Denoised MDP [28] only model *controllable* and reward-relevant aspects of the environment. While simplified latents can make transition dynamics more tractable to model, they do not necessarily give rise to the systematic action representations that we are interested in. Lastly, Self Predictive Representation (SPR) models [7, 6, 5] learn dynamics by predicting the future representations of a target encoder. SPR models have been used both for model-free and model-based control.

Mutual information minimization is used in many deep learning frameworks more generally: [29] and [30] use variational methods for minimizing the mutual information between the network's inputs $X$ and latent representations $Z$ while maximizing the mutual information between representations $Z$ and outputs $Y$. Mutual information minimization also has links to generalization ability [31, 32, 33, 34, 35], robustness in RL [36, 37], and exploration [38, 39]. Our information bottleneck differs from previous approaches in that it directly constrains the effect the latent state can have on the residual term in the latent dynamics over and above the agent's actions.

Closest to our regularization method is the *past-future* information bottleneck [40, 41]. Here the mutual information between sequences of past states and future states is minimized [42, 43]. While this method simplifies dynamics, our approach differs in important ways: Rather than representing the environment's dynamics, say, using a low number of its principal components, we treat the dynamics operator $\tilde{\Delta}_t$ itself as a random variable, and minimize its conditional dependence on $\mathbf{z}_t$. Furthermore, when $\tilde{\Delta}_t$ is fully disentangled from $\mathbf{z}_t$, each action can be seen as a transformation that acts on the latent state space in the same way, invariantly of $\mathbf{z}_t$ [44, 45]. We model the dynamics as *softly* state-invariant, allowing us to predict future latents both accurately and parsimoniously.

## 6   Conclusion

We have proposed a world model that tries to represent the effect of actions parsimoniously. Our model predicts future states while minimizing the dependence between the predicted dynamics $\tilde{\Delta}_t$ and the latent state representations $\mathbf{z}_t$. Optimizing this objective makes the effect of the actions on the agent's latent state more predictable. Combining our objective with different model classes – contrastive world models and SPR models – we observed consistent improvements in models' ability

to predict their own representations accurately, generalize to novel and noisy environments, and perform planning and model-free control in high-dimensional and pixel-based environments with complex dynamics. Overall, our results suggest that systematic action representations can offer important improvements to the generalization ability and data-efficiency of world models.

**Limitations:** Our model, in its current formulation, assumes that actions have predictable and deterministic effects on the environment. Aspects of the environment that do not behave predictably conditioning on actions are susceptible to be ignored by the model, even if they are relevant for the downstream task. While performance in Atari was improved on average, this aspect led to reduced performance in some tasks. Similarly, the degree of regularization $\beta$ needs to be tuned to achieve the right level of dynamics compression for different environments.

**Future directions:** A promising future direction is to combine our mutual information regularization with recurrent models. In partially observable, non-Markovian environments, next-state prediction is often done with a recurrent model that uses the agent's history as input. Using our bottleneck here would correspond to the assumption that the effect of the agent's actions are softly *history*-invariant. Another promising avenue of research is to combine PLSM with discrete dynamics. Many successful world modelling approaches assume that the latent state space is categorical [25, 46]. Instead of modelling transitions using affine transformations $\Delta$, transitions could be modelled by predicting transition matrices. Finally, recent recent advances in controllable video-generation like Genie [47] and UniSim [48] both incorporate actions into their video models. Genie infers *latent actions* that explain transitions in videos, and UniSim conditions on actions and language instructions to generate controllable videos. Modelling the effect of actions in a parsimonious way could improve the accuracy and generation capabilities of such systems.

## Acknowledgments

We thank the HCAI lab for valuable feedback and comments throughout the project. We are especially grateful for feedback from Luca Schulze Buschoff and Can Demircan on an earlier version of the manuscript, and indebted to Julian Coda-Forno for helping us name the model. This work was supported by the Institute for Human-Centered AI at the Helmholtz Center for Computational Health, the Volkswagen Foundation, the Max Planck Society, the German Federal Ministry of Education and Research (BMBF): Tübingen AI Center, FKZ: 01IS18039A, and funded by the Deutsche Forschungsgemeinschaft (DFG, German Research Foundation) under Germany's Excellence Strategy–EXC2064/1–390727645.15/18.

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

# Appendix

## A  Mutual Information minimization

To simplify dynamics, our algorithm minimizes the mutual information between $\mathbf{z}_t$ and $\tilde{\Delta}_t$ conditioned on $\mathbf{a}_t$ (Eq. 8 & 3). With stochastic gradient descent, we search for parameters $\theta$ that optimize the following objective:

$$\min_\theta I(\mathbf{z}_t; \tilde{\Delta}_t | \mathbf{a}_t) \tag{10}$$

To do so, we introduce a new variable $\mathbf{h}_t$ that depends on $\mathbf{z}_t$ and $\mathbf{a}_t$ (Eq. 6). We use this variable, together with $\mathbf{a}_t$, to produce $\tilde{\Delta}_t$, instead of $\mathbf{z}_t$. For a specific $\mathbf{a}_t$, the Markov chain behind $\tilde{\Delta}_t^{\mathbf{a}_t}$ can be written as

$$\mathbf{z}_t \to \mathbf{h}_t \to \tilde{\Delta}_t^{\mathbf{a}_t} \tag{11}$$

Thus, to minimize the mutual information between $\mathbf{z}_t$ and $\tilde{\Delta}_t$, we can instead minimize the mutual information between $\mathbf{z}_t$ and $\mathbf{h}_t$: $\min_\theta I(\mathbf{z}_t; \mathbf{h}_t | \mathbf{a}_t)$. Following [29], we write the mutual information between $\mathbf{z}_t$ and $\mathbf{h}_t$ as follows:

$$I(\mathbf{z}_t; \mathbf{h}_t | \mathbf{a}_t) = \int d\mathbf{z}\, d\mathbf{h}\, d\mathbf{a}\, p(\mathbf{z}, \mathbf{h}, \mathbf{a}) \log \frac{p(\mathbf{h} | \mathbf{z}, \mathbf{a})}{p(\mathbf{h} | \mathbf{a})} \tag{12}$$

In our case, $p(\mathbf{h} | \mathbf{z}, \mathbf{a})$ is deterministic. The marginal distribution conditioned on $\mathbf{a}$, $p(\mathbf{h} | \mathbf{a})$, is challenging to obtain. We instead use a variational approximation: With our variational distribution $q(\mathbf{h} | \mathbf{a})$, we can upper bound the mutual information as follows

$$I(\mathbf{z}_t; \mathbf{h}_t | \mathbf{a}_t) = \int d\mathbf{z}\, d\mathbf{h}\, d\mathbf{a}\, p(\mathbf{z}, \mathbf{a}) p(\mathbf{h} | \mathbf{z}, \mathbf{a}) \log \frac{p(\mathbf{h} | \mathbf{z}, \mathbf{a})}{q(\mathbf{h} | \mathbf{a})} \tag{13}$$

First, this upper bound lets us see that the mutual information can be minimized by minimizing the number of bits of information $\mathbf{z}_t$ contains about $\mathbf{h}_t$ over and above $\mathbf{a}_t$. This could potentially be done by introducing a parameterized prior that depends on $\mathbf{a}_t$, $q_\theta(\mathbf{h}_t | \mathbf{a}_t)$ and minimize the KL divergence between these two probability distributions

$$KL_D[p(\mathbf{h}_t | \mathbf{z}_t, \mathbf{a}_t) || q_\theta(\mathbf{h}_t | \mathbf{a}_t)] \tag{14}$$

We opt for a simpler approach, that allows us to do this without introducing a new prior and parameterization. Following [8], we assume that $q(\mathbf{h}_t | \mathbf{a}_t)$ is a standard, $d$-dimensional, isotropic Gaussian distribution $\mathcal{N}(\mathbf{0}, \mathbb{I})$. Assuming further that $\mathbf{h}_t$ is another $d$-dimensional isotropic Gaussian, the KL divergence is a function of the mean and standard deviation of $\mathbf{h}_t$ [8, 27]:

$$2KL_D = ||\mu(\mathbf{h}_t)||_2^2 + d + \sum_i^d \sigma(\mathbf{h}_t)_i - \log \sigma(\mathbf{h}_t)_i \tag{15}$$

Here $\mu(\cdot)$ and $\sigma(\cdot)$ return the mean and standard deviation of the Gaussian, respectively. Importantly, we can minimize the KL divergence by minimizing the first term in the above equation, which is simply the norm of the mean of $\mathbf{h}_t$. In our deterministic setup, we therefore simply minimize the norm of $\mathbf{h}_t$ itself.

## B  PLSM vs $L_1$ and $L_2$ norm regularization

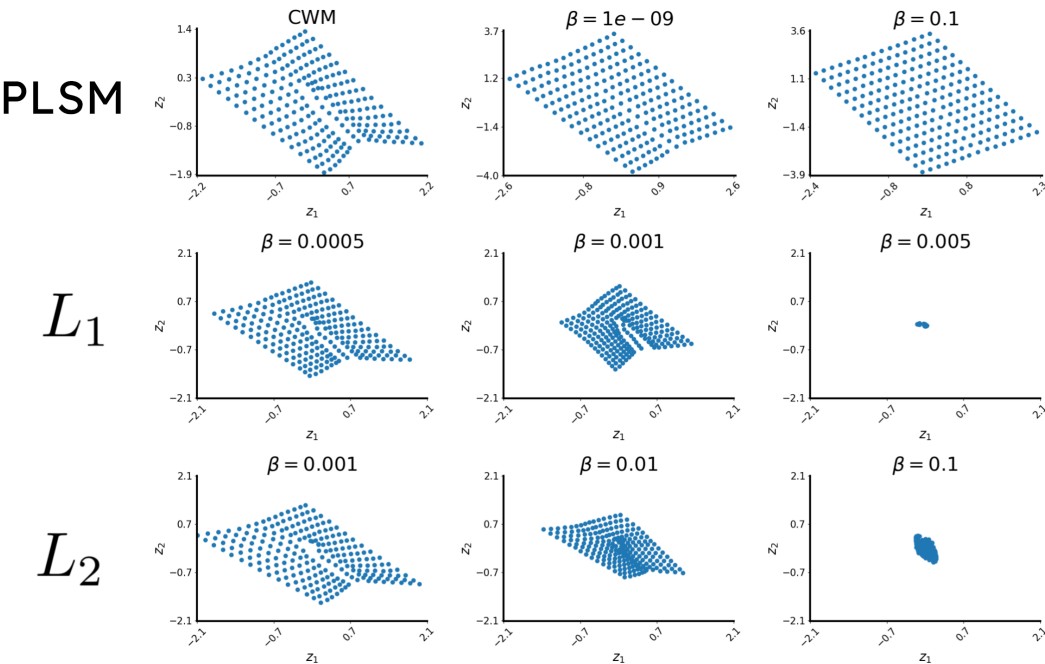

Figure 8: To illustrate how our mutual information minimization impacts the latent space, we trained PLSM to learn the dynamics of a dot moving in four directions on a gridworld with a wall in the middle. We compared PLSM to contrastive models that regularize the $L_1$ and $L_2$ norms of the latent space, respectively. We observe that PLSM regularization leads to a more regular representation of the states in the gridworld, whereas no regularization leads to a warped representation. Moreover, $L_1$ and $L_2$ regularization simply shrinks the latent space. This type of shrinkage is not present in PLSM.

## C  Ablations

We tested some alternative formulations and ablations of our PLSM objective on a subset of the datasets (see Fig. 9). One ablation removed the query representation $\mathbf{h}_t$ and simply regularized the latent state $\mathbf{z}_t$. This model achieved substantially worse performance in the three datasets we tested it on. As an alternative to regularizing the $L_2$ norm of $\mathbf{h}_t$, we also tested a *top-k* bottleneck, which used the top 15 features from $\mathbf{h}_t$ instead of the full representation. This model performed on par with the original PLSM. Lastly, minimizing the $L_2$ norm of $\mathbf{h}_t$ could potentially be compensated for by the dynamics MLP by increasing the norm of the weight matrices. As a control, we added weight decay to the dynamics model to prevent this, and observe comparable performance.

## D  Is the $L_2$ minimization effective?

One potential issue with minimizing the $L_2$ norm of the query representation $\mathbf{h}_t$ is that the model can compensate for this by increasing the magnitude of the weights in the ensuing dynamics module. While this can be prevented by adding weight decay to the dynamics model's weights, we show that PLSM does not suffer from this empirically in the datasets we evaluated it on. We calculated the average norm of $\mathbf{h}_t$ of PLSM relative to the average norm of $\mathbf{z}_t$ in the unregularized model. Comparing them we see indeed that $\mathbf{h}_t$ has several orders of magnitude lower norm. However, comparing the norm of the ensuing linear layers, we see that they most often do not differ significantly (see Fig. 10). This suggests that shrinking $\mathbf{h}_t$ truly minimizes the amount of information the model can extract from it to predict $\Delta$.

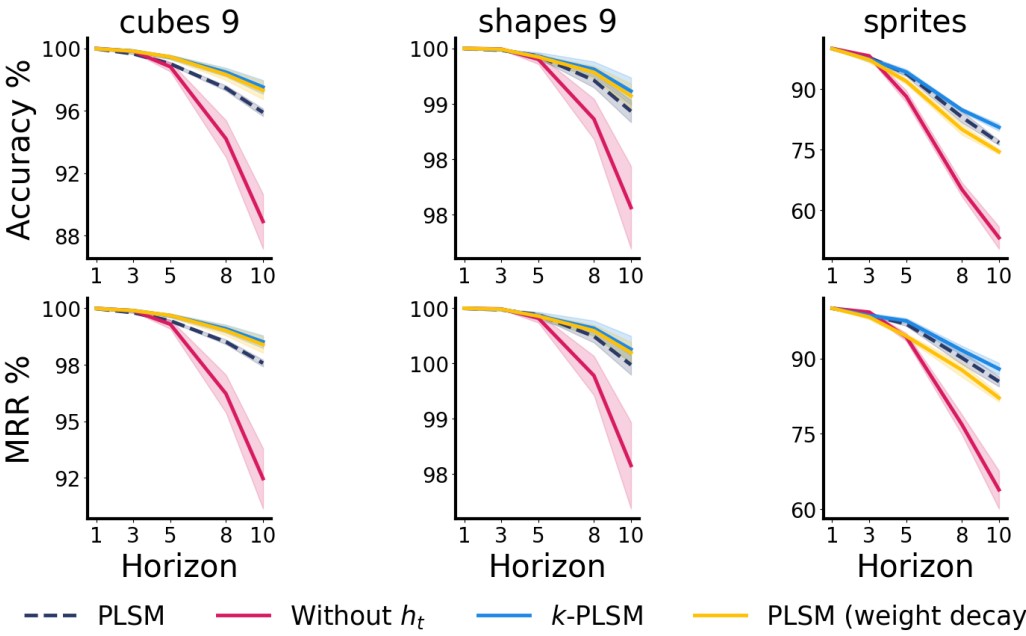

Figure 9: PLSM and three alternative models, one dispensing of the query representation, one using a top-$k$ bottleneck, and one adding weight decay to the dynamics MLP. Removing the query representation decreases performance substantially. Performance is intact in the two alternative formulations of PLSM.

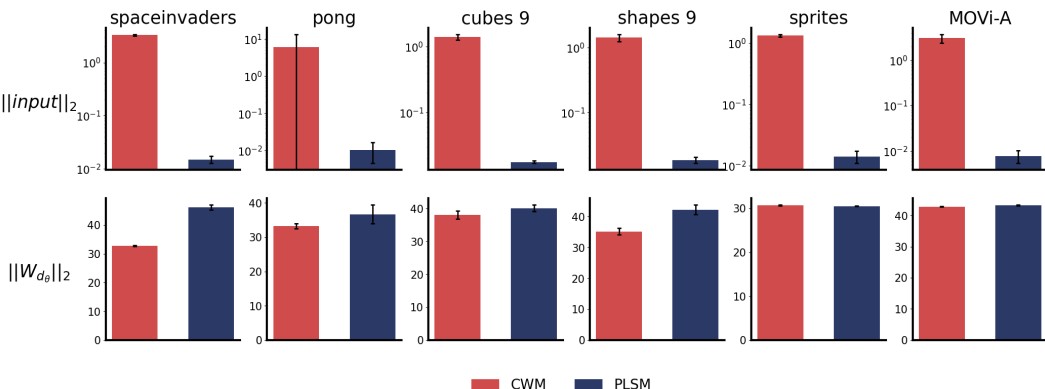

Figure 10: While the norm of $\mathbf{h}_t$ is orders of magnitude lower than the norm of $\mathbf{z}_t$, the norm of ensuing weights from the dynamics model are comparable and often not significantly different.

# E   Atari

We report more detailed Atari results in this section, including median, IQM, and the probability of improvement using the RLiable package [17] (see Fig. 11) . We also investigated how the regularization strenght of PLSM impacted performance in Atari. We see that in games where important features of the environment are not controllable by the agent, weaker regularization is beneficial. For instance, in Boxing, the movement of the opponent is hard to predict, and here having a lower regularization strength is advantageous.

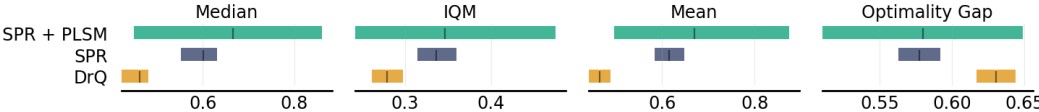

Figure 11: Median, IQM, Mean and Optimality Gap reported for the three model-free RL algorithms.

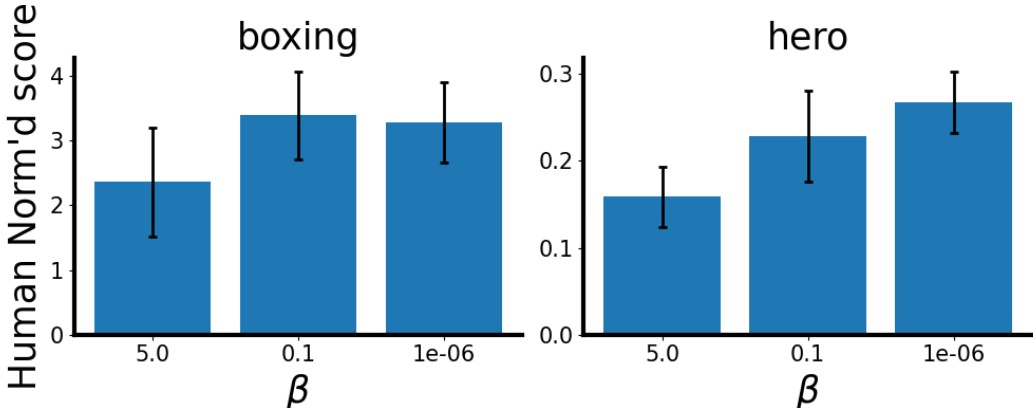

Figure 12: In Boxing and Hero, lower regularization generally led to better performance.

## F   Datasets

See Fig. 13 for example frames from the training datasets and Fig. 14 for example frames from the generalization tests.

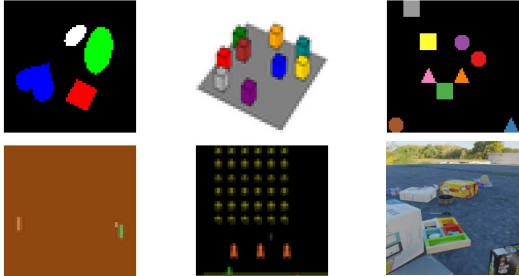

Figure 13: The six environments used to evaluate the accuracy of the latent dynamics. Top, from left to right: dSprites 4, cubes 9m shapes 9. Bottom: pong, space invaders, and MOVi-e.

## G   Hybrid model

We propose a hybrid model for the Atari environments. The hybrid model splits the latent state in two, one parsimonious state space $\mathbf{z}_t^1$ and an unconstrained state space $\mathbf{z}_t^2$. To predict the next state, the model uses two dynamics MLPs $\tilde{\mathbf{z}}_{t+1}^2 = \mathbf{z}_t^2 + d_\theta^1(\mathbf{z}_t^1, \mathbf{z}_t^2, \mathbf{a}_t)$ and $\tilde{\mathbf{z}}_{t+1}^1 = \mathbf{z}_t^1 + d_\theta^2(\mathbf{h}_t, \mathbf{a}_t)$. The next latent prediction is then defined as $\tilde{\mathbf{z}}_{t+1} = Concatenate(\tilde{\mathbf{z}}_{t+1}^1, \tilde{\mathbf{z}}_{t+1}^2)$. Results are shown in Fig. 15.

## H   Slots with information regularization

We evaluated the C-SWM on our cubes 9 task, both with and without our regularization. The performance of C-SWM dropper by a small amount due to the increased difficulty of the task for

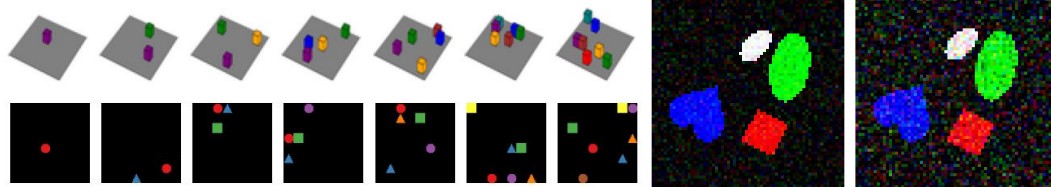

Figure 14: Datasets used to probe generalization and robustness. **Left**: Cubes and shapes data with varying numbers of objects. **Right**: dSprite images with mild and severe noise corruption.

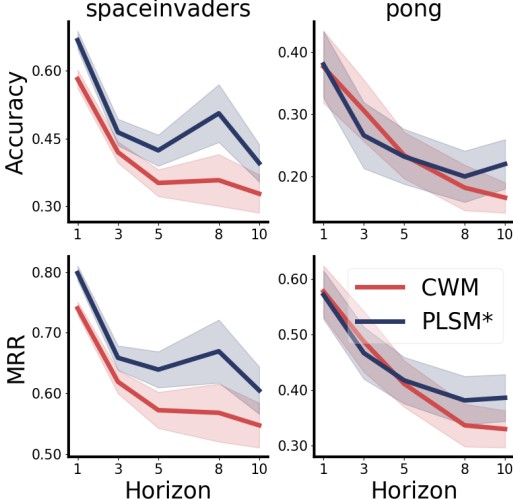

Figure 15: Our hybrid model, which splits the latent space into a parsimonious one and an unconstrained one, outperforms the baseline on long-horizon latent prediction in the Atari environments.

longer prediction horizons ($t = 10$). Interestingly, using our regularization method proved beneficial for the slotted model, outperforming the unregularized C-SWM in long horizon latent prediction.

Table 1: PLSM with slots outperforms the C-SWM [3] in latent prediction at prediction horizon $t = 10$, averaged over five seeds, plus and minus standard error of the mean.

| Accuracy | PLSM | C-SWM |
|---|---|---|
| Cubes 9 | **99.05**% $\pm 0.18$ | 97.45%$\pm 1.3$ |

# I Atari scores

| Game | Random | Human | CURL | DrQ | SPR | SPR+PLSM |
|------|--------|-------|------|-----|-----|----------|
| Alien | 227.8 | 7127.7 | 711.0 | 865.2 | **841.9** | 832.2 |
| Amidar | 5.8 | 1719.5 | 113.7 | 137.8 | **179.7** | 152.1 |
| Assault | 222.4 | 742.0 | 500.9 | 579.6 | 565.6 | **634.5** |
| Asterix | 210.0 | 8503.3 | 567.2 | 763.6 | **962.5** | 913.4 |
| BankHeist | 14.2 | 753.1 | 65.3 | 232.9 | **345.4** | 63.5 |
| BattleZone | 2360.0 | 37187.5 | 8997.8 | 10165.3 | **14834.1** | 13394.0 |
| Boxing | 0.1 | 12.1 | 0.9 | 9.0 | **35.7** | 28.4 |
| Breakout | 1.7 | 30.5 | 2.6 | 19.8 | **19.6** | 15.6 |
| ChopperCommand | 811.0 | 7387.8 | 783.5 | 844.6 | **946.3** | 450.6 |
| CrazyClimber | 10780.5 | 35829.4 | 9154.4 | 21539.0 | **36700.5** | 30410.2 |
| DemonAttack | 152.1 | 1971.0 | 646.5 | 1321.5 | **517.6** | 477.2 |
| Freeway | 0.0 | 29.6 | 28.3 | 20.3 | 19.3 | **28.4** |
| Frostbite | 65.2 | 4334.7 | 1226.5 | 1014.2 | 1170.7 | **1371.9** |
| Gopher | 257.6 | 2412.5 | 400.9 | 621.6 | **660.6** | 557.0 |
| Hero | 1027.0 | 30826.4 | 4987.7 | 4167.9 | **5858.6** | 5763.2 |
| Jamesbond | 29.0 | 302.8 | 331.0 | 349.1 | **366.5** | 336.8 |
| Kangaroo | 52.0 | 3035.0 | 740.2 | 1088.4 | 3617.4 | **4886.6** |
| Krull | 1598.0 | 2665.5 | 3049.2 | 4402.1 | 3681.6 | **4043.4** |
| KungFuMaster | 258.5 | 22736.3 | 8155.6 | 11467.4 | 14783.2 | **15187.6** |
| MsPacman | 307.3 | 6951.6 | 1064.0 | 1218.1 | **1318.4** | 1156.7 |
| Pong | -20.7 | 14.6 | -18.5 | -9.1 | -5.4 | **1.1** |
| PrivateEye | 24.9 | 69571.3 | 81.9 | 3.5 | **86.0** | 85.8 |
| Qbert | 163.9 | 13455.0 | 727.0 | 1810.7 | **866.3** | 786.5 |
| RoadRunner | 11.5 | 7845.0 | 5006.1 | 11211.4 | 12213.1 | **12400.0** |
| Seaquest | 68.4 | 42054.7 | 315.2 | 352.3 | 558.1 | **561.3** |
| UpNDown | 533.4 | 11693.2 | 2646.4 | 4324.5 | 10859.2 | **29572.0** |
| #Superhuman | 0 | N/A | 2 | 3 | **6** | **6** |
| Mean | 0.0 | 1.000 | 0.261 | 0.465 | 0.616 | **0.671** |

# J  Hyperparameters

## J.1  Compute

We ran all experiments reported in the paper on compute nodes with 2 Nvidia A100 GPUs. On average, Atari runs lasted for 5 hours and DMC runs 4-9 hours depending on the task.

## J.2  Convolutional neural network architecture

To learn pixel representations we use a Convolutional Neural Network (CNN) architecture similar to the one used in [9] and [49]. For the Atari latent prediction experiments we stacked two frames to provide information about object movements. In MOVi-E we stacked four frames.

```python
import torch
from torch import nn
encoder  =  nn.Sequential(
                nn.Conv2d(num_channels, 32, 3, stride=2),
                nn.ReLU(),

                nn.Conv2d(32, 32, 3, stride=1),
                nn.ReLU(),

                nn.Conv2d(32, 32, 3, stride=1),
                nn.ReLU(),

                nn.Conv2d(32, 32, 3, stride=1),
                nn.ReLU()
                )
```

## J.3  Contrastive models

Table 2: Contrastive model hyperparameters.

| Hyperparameter | Value |
|---|---|
| Hidden units | 512 |
| Batch size | 512 |
| MLP hidden layers | 2 |
| Latent dimensions $|\mathbf{z}_t|$ | 50 |
| Query dimensions $|\mathbf{h}_t|$ | 50 |
| Regularization coefficient $\beta$ | 0.1 |
| Margin $\lambda$ | 1 |
| Learning rate | 0.001 |
| Activation function | ReLU [50] |
| Optimizer | Adam [51] |

## J.4  DeepMind Control Suite

We use the same hyperparameters as [6] for the planning agents. The only addition is our PLSM regularization. We use a regularization coefficient of $0.1$ for all agents, and make the query net MLP $f_\theta$ the same size as the dynamics network $\mathbf{h}_t$, with $|\mathbf{h}_t| = |\mathbf{z}_t|$.

Table 3: Action repeat values in the DeepMind Control Suite tasks.

| Task | Action repeat |
|---|---|
| Acrobot Swingup | 4 |
| Finger Turn Hard | 2 |
| Quadruped Walk | 4 |
| Quadruped Run | 4 |
| Humanoid Walk | 2 |

## J.5 Distracting Control Suite

All models were trained with an action repeat value of 2 in all environments. All model hyperparameters were the same is in [12]. The background videos were randomly sampled per episode from the DAVIS 2017 video dataset [52], projected in black and white.

