# OpenReview forum: "Simplifying Latent Dynamics with Softly State-Invariant World Models"
_NeurIPS.cc/2024/Conference — NeurIPS 2024 poster_

### Official Review · Reviewer_VZsj · 2024-06-21

**Soundness:** 3
**Presentation:** 3
**Contribution:** 3
**Rating:** 6
**Confidence:** 4

**Summary:**

This paper proposes a Parsimonious Latent Space Model (PLSM) as a latent world model method. The main idea of PLSM is twofold: (1) the use of an additional hidden variable $h_t$, which is bottlenecked to have parsimonious information (for better latent predictability) and (2) the use of difference prediction instead of latent prediction. The authors combine PLSM with different world model/RL techniques (CWM, TD-MPC, and SPR), and show that it leads to better long-term prediction and improves performance on RL control and game benchmarks.

**Strengths:**

* Generally the paper is well-written, and I enjoyed reading the draft.
* The related work section extensively discusses the difference between PLSM and previous approaches.
* The proposed method is relatively easy to implement.
* The authors experimentally show that PLSM improves long-time prediction accuracy.
* The experiments are done across three very different settings (long-term prediction, model-based RL, and model-based RL).

**Weaknesses:**

* I'm not sure how simply minimizing the norm of $h_t$ leads to minimal representations. If $h_t$ were a stochastic random variable with a fixed variance, this would make sense (since in this case, this becomes equivalent to KL divergence minimization between the posterior and the standard Gaussian prior). However, $h_t$ seems deterministic in this work, and I don't think just minimizing $\\\|h_t\\\|$ has a principled regularization effect, given that it is possible to contain the same amount of information with a very small scale (e.g., multiplying $h_t$ by $10^{-9}$ only makes the norm smaller while maintaining the same amount of information).
* I'm also not fully convinced that we *need* an additional $h_t$ variable. Why can't we just impose a bottleneck on $z_t$? Is there a unique benefit of having an additional latent $h_t$ variable that is not achievable by having $z_t$ alone?
* While one of the emphasized contributions seems to be the fact that PLSM makes difference prediction ($z_{t+1}$ - $z_t$) instead of latent prediction ($z_{t+1}$), I believe this is an extremely common trick in model-based RL, which I even consider to be the "default" design choice in model-based RL.
* The performance gain over previous RL methods (TD-MPC, SPR) seems marginal, despite the added complexity of PLSM. Moreover, the empirical comparisons are weak in the sense that the comparison is only made with a single method in each setting, and it's unclear how PLSM compares to other regularization methods (e.g., RePo, other regularization techniques, etc.).

In summary, I feel this is a borderline paper. The proposed method (as well as its writing) is clean and conceptually intriguing, but its novelty and performance gain seem marginal. I'd recommend a borderline reject at this point, but I'm willing to adjust the score if the authors address the concerns above (especially the second and fourth points).

**Questions:**

Please see the weaknesses section above.

**Limitations:**

The authors sufficiently address the limitations in Section 7.

---

> ### Author Rebuttal · Authors · 2024-08-06
>
> We would like to thank the reviewer for the thoughtful review and we are glad that the reviewer enjoyed reading our paper. The reviewer raised several important points regarding the effect of our regularization and how it compares to other regularization methods in RL. In response. we show that our regularization indeed has a principled effect (see Supporting Figure D), that PLSM shows considerable improvements over another robust baseline, RePO, in the Distracting DeepMind Control Suite (see Supporting Figure A), and lastly that regularizing a separate query representation $h_t$ provides a substantial improvement over regularizing $z_t$ (see Supporting Figure F). We believe that addressing the concerns raised by the reviewer has helped us improve our paper substantially. We provide detailed responses to each question below:
>
> >I'm not sure how simply minimizing the norm $h_t$ of leads to minimal representations
>
> We thank the reviewer (along with reviewer 2) for bringing this criticism to our attention. We investigated whether PLSM learns higher norm weight matrices to compensate for the low norm $h_t$. First, we found that the norm of PLSM’s dynamics weights were not substantially larger than those of the baseline model, whereas the norm of $h_t$ often was several orders of magnitude smaller than $z_t$ (see Supporting Figure D). Additionally, we show in Figure 5 in the paper that the query representations carry less information about the generative factors than the latent representations. Lastly, even though our model does not compensate with large weights, we propose two methods for preventing this. i) by penalizing the L2 norm of the dynamics weights, and ii) by only using the top-k largest features in the query representation. We trained these two approaches on a subset of the datasets and found that they perform comparably, and sometimes better than the original PLSM formulation (see Supporting Figure F).
>
> >I'm also not fully convinced that we need an additional variable $h_t$. Why can't we just impose a bottleneck on $z_t$?
>
> To address this point we evaluated a version that imposes an information bottleneck directly on $z_t$, instead of the query representation $h_t$. We observed a general reduction in performance on the three datasets tested on (See Supporting Figure F), suggesting that $h_t$ is necessary. The advantage of regularizing the query representation is that we only encourage the model to compress away aspects of the environment whose dynamics are unpredictable. In an environment with easily predictable dynamics, our method won’t necessarily lead to any loss of information, whereas imposing a bottleneck on z will invariably do that.
>
> >one of the emphasized contributions seems to be the fact that PLSM makes difference prediction ($z_{t+1} - z_t$) instead of latent prediction ($z_{t+1}$), I believe this is an extremely common trick in model-based RL
>
> We agree with the reviewer that this is common practice in model-based RL. We did not intend to make this sound like a key contribution in our paper, but we emphasized this modeling choice because our regularization does not make much sense without it. We have updated the introduction to reflect this:
>
> **"As is common practice in many dynamics model architectures [1, 2], we consider the case where the model predicts the next latent $\hat{z}_{t+1}$ state by predicting the *difference* $\Delta$, or the *change*, between the current and future latent state, given an action $a_t$."**
>
> >it's unclear how PLSM compares to other regularization methods (e.g., RePo, other regularization techniques, etc.)
>
> We followed the reviewer’s suggestion and we evaluated PLSM against another robust model-based RL algorithm, RePO [3], in five Distracting DMC environments [4]. In these environments, the background was replaced with a distracting video from the DAVIS 2017 dataset. RePO uses a GRU-based dynamics model, akin to Dreamer, but without pixel reconstruction. We incorporated our regularization into RePO by learning a compressed query representation, and passed this alongside the action to the GRU, making the GRU dynamics softly state-invariant. We observe considerable performance gains in the more challenging environments like hopper-stand, finger-spin and walker-run, suggesting that our regularization can be beneficial together with methods designed for robustness (see Supporting Figure A, scores are averaged across 5 seeds, with standard error shown as shaded region).
>
> References:
>
> [1] PILCO: A Model-Based and Data-Efficient Approach to Policy Search, Deisenroth et al., ICML 2011
>
> [2] Data efficient Reinforcement Learning With Self-Predictive Representations, Schwarzer et al., ICLR 2021
>
> [3] RePo: Resilient Model-Based Reinforcement Learning by Regularizing Posterior Predictability, Zhu et al., NeurIPS 2023
>
> [4] The Distracting Control Suite – A Challenging Benchmark for Reinforcement Learning from Pixels, Stone et al., ArXiv, 2021

---

> > ### Comment · Reviewer_VZsj · 2024-08-07
> > **Thanks for the response**
> >
> > Thanks for the detailed response.
> >
> > While I appreciate the new results about $\\\|h\\\|$, I don't think the fact that the norm of weights ($\\\|w\\\|$) remains similar supports the effect of the norm regularizer (why is it related to that?). Moreover, even if it does support it, this finding is at best purely empirical and not principled (by "principled", I mean having a theoretical or logical justification), given that it *does* have the theoretical issue that the norm of $h$ can be arbitrarily small while not losing any information. That being said, this was not a major concern for me, so I'd be fine if the authors clearly acknowledge that this is an *empirically* motivated regularizer in the paper with sufficient discussion.
> >
> > I think I still didn't fully understand the necessity of $h$. In Figure 1, why can't we just merge $z_1$ and $h_1$, $z_2$ and $h_2$, and so on? How can imposing an information bottleneck only on $h$ prevent information loss, given that there's no direct path from (say) $z_1$ to $z_2$ in Figure 1? (i.e., it must pass through $h_1$, so the bottleneck on $h_1$ will anyway cause information loss toward $z_2$ --- or is this figure somewhat incorrect?) I can see that having an additional $h$ leads to *empirical* performance improvement, but I'm curious if there are any *principled* benefits of additionally having $h$, or if it is purely a practical design choice. I might have missed something, and could the authors elaborate on this point?

---

> > > ### Author Response · Authors · 2024-08-07
> > >
> > > We thank the reviewer for the follow-up questions. The motivation for minimizing the L2 norm comes from [1], where they argue that it constrains the size of the latent space, and is therefore a suitable regularizer. They also show that minimizing the L2 norm amounts to minimizing the KL divergence if the prior is a constant variance Gaussian with zero mean. Here they also propose to regularize the L2 norm of the ensuing weights, to avoid unbounded optimization. We therefore show that our regularization does constrain the size of the latent space while the dynamics weights norms stay similar. The concern about the norm of the dynamics weights was also raised by reviewer gEcM. Note that we also show that our method works experimentally by sparsifying $h_t$ to only include the largest $k$ features (reducing from 50 to 15 in our case, see Supporting Figure E).
> > >
> > > Still, we agree with the reviewer that the argument presented in the paper is empirical. We have therefore added the following paragraph in the methods section to highlight this.
> > >
> > > **”We minimize the $L_2$ norm of $h_t$ in order to constrain the size of the query representation. This type of regularization has been used to regularize representations of deterministic Autoencoders in past work [1, 2], with [1] showing that it is equivalent to minimizing the KL divergence to a constant variance zero mean Gaussian. Other regularizers are also possible. Instead of modeling $h_t$ as a deterministic variable, it can be treated stochastically. Using multivariate Gaussian distributions with learned variances, one can regularize $h_t$ by minimizing the KL divergence to a standard Gaussian [3]. For simplicity we use the deterministic variant and leave stochastic versions for future work.”**
> > >
> > > >How can imposing an information bottleneck only on $h$ prevent information loss, given that there's no direct path from (say) $z_1$ to $z_2$ in Figure 1? (i.e., it must pass through $h_1$, so the bottleneck on $h_1$ will anyway cause information loss toward $z_2$)
> > >
> > > This is a good question. The reason why passing $z_t$ through the $h_t$ bottleneck does not necessarily cause information loss is because $h_1$ is not used to predict $z_2$ directly, but rather the *change* term $\Delta$. As such, the model is designed to have information loss with respect to $\Delta$, but not necessarily $z_{t+1}$. If $z$ evolves in a manner that is independent of $z$, the model can ignore it to predict $\Delta$, without having to lose any information contained in $z$. We have added the following clarification to the model section, and are happy to.
> > >
> > > **”Regularizing $h_t$ differs from regularizing $z_t$ in important ways. In environments where the dynamics $\Delta$ can be predicted perfectly from the actions and independently of the state, regularizing $h_t$ will not lead to a loss in information in $z_t$. This is because the bottleneck on $h_t$ only constrains the model in using information from $z_t$ to predict $\Delta$, and not necessarily in predicting $z_{t+1}$.”**
> > >
> > > References:
> > >
> > > [1] From Variational to Deterministic Autoencoders, Ghosh et al., ICLR 2020
> > >
> > > [2] Improving Sample Efficiency in Model-Free Reinforcement Learning from Images, Yarats et al., ArXiv 2020
> > >
> > > [3] Auto-Encoding variational Bayes, Kingma et al., ICLR 2014

---

> > > > ### Comment · Reviewer_VZsj · 2024-08-07
> > > > **Thanks for the response**
> > > >
> > > > Thanks for the response.
> > > >
> > > > For the first point, yes, as I also mentioned in my initial response, if $\|h\|$ were a stochastic variable, norm regularization would be somewhat equivalent to KL minimization to a Gaussian prior. I can also imagine that it may have a similar effect even for deterministic $h$'s in practice. I believe the new clarification would help prevent this kind of confusion.
> > > >
> > > > For the second point, thanks for the clarification. I got the point. Perhaps the main issue was then that Figure 1 is somewhat misleading, because it seems there's no direct flow from $z_1$ to $z_2$ (even though there is --- $z_2$ is the sum of $z_1$ and the prediction from $h_1$ and $a_1$). I would encourage the authors to revise Figure 1 to clarify this point.
> > > >
> > > > In general, my main concerns have mostly been resolved to some degree. I also appreciate new preliminary results with additional baselines. Accordingly, I've adjusted my score to 6.

---

> ### Author Response · Authors · 2024-08-07
> **Thanks for reviewing our work**
>
> We thank the reviewer for bringing these points to our attention. We believe that addressing them has substantially improved our paper, and we are grateful that the reviewer raised the score accordingly. We will also revise Figure 1 to clarify that the dynamics model predicts $\Delta$. Thank you again!

---

### Official Review · Reviewer_hbwZ · 2024-07-09

**Soundness:** 2
**Presentation:** 3
**Contribution:** 2
**Rating:** 5
**Confidence:** 4

**Summary:**

This paper presents an information bottleneck principle to regularize the latent dynamics, which makes the effect of the agent’s actions more predictable. This approach minimizes the mutual information between latent states and the change that action produces in the agent’s latent state, in turn minimizing the dependence the state has on the dynamics. The proposed world model can be combined with model-based planning and model-free RL.

**Strengths:**

The proposed method reasonably controls the information contained in the latent space, regularizing the behavior of the latent space dynamic model and allowing for strong generalization.

**Weaknesses:**

- The contribution is limited since the MI regularization for RL representation has been adopted in several previous works, including [1-3].

- The empirical results presented in this paper do not show significant performance improvements.


[1] Learning Robust Representations via Multi-View Information Bottleneck. ICLR 2020

[2] Dynamic Bottleneck for Robust Self-Supervised Exploration. NeurIPS 2021

[3] InfoBot: Structured Exploration in ReinforcementLearning Using Information Bottleneck. ICLR 2019

**Questions:**

- Since the MI between representation and state is constrained, should the algorithm exhibit stronger robustness in various domains? Can you show some empirical results?

---

> ### Author Rebuttal · Authors · 2024-08-06
>
> We thank the reviewer for the constructive feedback and for the helpful suggestions. To address the concerns raised in the review, we have added another set of experiments showing that PLSM improves robustness in visual control tasks with distracting background videos (see Supporting Figure A), as well as improving our results on the Atari control suite by tuning the regularization strength parameter (see Supporting Figure C). We agree with the reviewer that mutual information (MI) based regularizers themselves are not novel in RL. However, our particular MI method is novel, and aims at regularizing a part of latent dynamics models that to our knowledge has not been explored before. We extend the discussion on other MI based methods in RL in the related works section, including the papers cited above. In sum, we believe that addressing the points the reviewer outlined has made our paper better. We give detailed answers to the reviewer’s questions below:
>
> >The contribution is limited since the MI regularization for RL representation has been adopted in several previous works, including [1-3].
>
> We have extended the related works section with the following paragraph:
>
> **"Mutual information minimization methods have been effectively applied for representation learning in RL too. Previous works have focused on using information bottlenecks to improve generalization [1, 2, 3], robustness [4] and exploration [5, 6]. Our information bottleneck differs from these by constraining the effect the latent state can have on the residual term in the latent dynamics."**
>
>
> >Since the MI between representation and state is constrained, should the algorithm exhibit stronger robustness in various domains?
>
> We evaluated the robustness properties of PLSM across five Distracting DeepMind Control Suite environments (Distracting DMC). Distracting DMC consists of challenging visual control tasks where the background has been replaced with videos that are irrelevant for solving the task. We used RePO [7] as a baseline, and implemented the PLSM regularization by minimizing how much the latent state influences the dynamics of the GRU in RePOs world model, while still conditioning on actions. We observe considerable improvements in more challenging tasks (hopper-stand, finger-turn and walker-run) and smaller improvements in walker-walk (see Supporting Figure A). This suggests that PLSM really shines in cases where the agent needs to learn to compress away irrelevant information to generalize.
>
> >The empirical results presented in this paper do not show significant performance improvements.
>
> We have extended our evaluation of PLSM in Distracting DMC, and the standard DeepMind Control Suite. We also optimized our regularization strength in Atari and found that $\beta=5$ gave the best performance overall, improving human normalized score in Atari from 61.5 % for SPR, to 67% with PLSM (see Supporting Figure C). Note that SPR and DrQ interquartile ranges are lower since they are based on 100 seeds. In DMC, we see improvements in many domains, especially in challenging tasks like humanoid-walk (see Supporting Figure B). In the Distracting DMC tasks, we see larger performance gains as well.
>
> References:
>
> [1] Generalization and Regularization in DQN, Farebrother et al., ArXiv 2018
>
> [2] DeepMDP: Learning Continuous Latent Space Models for Representation Learning
>
> [3] Learning Robust Representations via Multi-View Information Bottleneck, Federici et al.,  ICLR 2020
>
> [4] Robust Predictable Control, Eysenbach et al., NeurIPS 2022
>
> [5] Dynamic Bottleneck for Robust Self-Supervised Exploration, Bai et al., NeurIPS 2021
>
> [6] InfoBot: Structured Exploration in ReinforcementLearning Using Information Bottleneck, Goyal et al., ICLR 2019
>
> [7] RePo: Resilient Model-Based Reinforcement Learning by Regularizing Posterior Predictability, Zhu et al., NeurIPS 2023

---

> > ### Comment · Reviewer_hbwZ · 2024-08-10
> > **Response**
> >
> > The additional experiments strengthen the contribution and I encourage you to perform complement experiments with multiple seeds in the next version.

---

> > > ### Author Response · Authors · 2024-08-11
> > >
> > > We thank the reviewer for engaging with our paper. We believe that addressing the reviewer's concerns has helped us improve our paper, and we are grateful the reviewer raised the score as a consequence.

---

### Official Review · Reviewer_itzz · 2024-07-12

**Soundness:** 3
**Presentation:** 3
**Contribution:** 3
**Rating:** 6
**Confidence:** 4

**Summary:**

This paper introduces a method to enforce parsimonious latent dynamic models. The key idea is that if we can minimise the influence of states on the dynamic, i.e. the conditional mutual information $I(z_t, \Delta_t | a_t)$, the dynamic can generalise better to unseen states during prediction. In order to achieve this goal, the author proposes PLSM based on information bottleneck. Experiments show that the proposed method can be widely applied to prediction, and both model-based and model-free RL. On specific tasks that fit well with the parsimonious assumption, PLSM significantly improve the result.

**Strengths:**

- The problem is well-motivated, and the solution is straightforward to implement.
- The experiments cover a rather broad range of problems which showcase the general applicability of the method.

**Weaknesses:**

- While PLSM exhibits strong performance on specific prediction tasks, the improvements in RL tasks are relatively modest.
- As is common with information bottleneck-based methods, PLSM's performance may be highly sensitive to the hyperparameter of the regularizer for the bottleneck.

**Questions:**

- Could you provide Interquartile Mean (IQM) results for the Atari experiments, as suggested by Agarwal et al. (2021) in "Deep Reinforcement Learning at the Edge of the Statistical Precipice"? This would offer a more robust measure of aggregate performance across multiple tasks.
- Since the regularisor weights $\beta$ is very important to the performance, I wonder how robust is PLSM to different $\beta$? Could you also share some insights about tuning the parameter? This can be a valuable information to the community.

**Limitations:**

Yes, the limitations are well-discussed by the authors.

---

> ### Author Rebuttal · Authors · 2024-08-06
>
> We thank the reviewer for their constructive feedback and helpful suggestions. We have taken steps to address the concern that performance improvements are relatively modest: We evaluated PLSM in environments with visual distractions and show that parsimonious dynamics can offer considerable performance gains (see Supporting Figure A). We have also performed an analysis in Atari where we investigate the sensitivity to regularization strength (see Supporting Figure F). In sum, we believe that adding these clarifications and showing that there are other settings where PLSM offers a greater advantage has considerably improved our paper, and we thank the reviewer again for suggesting these. We give more detailed answers to the reviewer’s questions below:
>
> >While PLSM exhibits strong performance on specific prediction tasks, the improvements in RL tasks are relatively modest.
>
> We evaluated PLSM in tasks where our regularization might yield a larger improvement. We chose the Distracting DeepMind Control Suite [1] for this, a challenging visual control environment where static backgrounds are replaced with distracting videos. Since PLSM seeks to compress away unpredictable aspects of the environment, we hypothesized that performance should improve in this setting. We evaluated PLSM against another robust model-based RL algorithm, RePO [2], in five tasks with distracting background videos. We incorporated our regularization into RePO by learning a compressed query representation, and passed this alongside the action to the GRU which RePO uses to model the latent dynamics. The GRU dynamics were therefore softly invariant of the latent state. We observe considerable performance gains in the more challenging environments like hopper-stand, finger-spin and walker-run, suggesting that our regularization is beneficial when the agent needs to learn to ignore irrelevant distractors (see Supporting Figure A, scores averaged across 5 seeds with standard error shown as shaded regions).
>
>
> >Could you provide Interquartile Mean (IQM) results for the Atari experiments, as suggested by Agarwal et al. (2021)?
>
> We optimized $\beta$ and found that $\beta = 5$ gave the strongest performance, improving mean human normalized score in Atari from 61.5 % for SPR, to 67% with PLSM. We now also report Interquartile Mean, Median and Mean Human normalized performance for this model. In all three metrics PLSM is advantageous (see Supporting Figure C). Note that SPR and DrQ interquartile ranges are lower since they contain scores from 100 seeds.
>
> >Since the regularisor weights $\beta$ is very important to the performance, I wonder how robust is PLSM to different $\beta$?
>
> After performing a more extensive hyperparameter search for Atari, we see that there are certain games that are more sensitive to the regularization strength than others. For instance, in tasks where there are important features that are unpredictable, like in Boxing and Hero, regularizing too much can be bad for performance (see Supporting Figure F). Still, we find two hyperparameter configurations that both beat the original SPR in terms of mean performance ($\beta = 5$ and $\beta = 0.1$). These two configurations only differ in mean performance by roughly 2 percentage points, indicating that performance is relatively stable on average with respect to $\beta$.
>
> References:
>
> [1] The Distracting Control Suite – A Challenging Benchmark for Reinforcement Learning from Pixels, Stone et al., ArXiv, 2021
>
> [2] RePo: Resilient Model-Based Reinforcement Learning by Regularizing Posterior Predictability, Zhu et al., NeurIPS 2023

---

> > ### Comment · Reviewer_itzz · 2024-08-09
> > **Thanks for your rebuttal, I decide to maintain my score.**
> >
> > Thank you for your rebuttal and the additional experimental results provided. While I appreciate the effort, I still have some concerns:
> >
> > - Regarding the Distracting DMC experiments: The comparison between RePO and RePO + PLSM is interesting, but may not fully demonstrate PLSM's effectiveness in handling distracting information. RePO already addresses this issue to some extent. A more illustrative comparison might be between Dreamer and Dreamer + PLSM, which could better showcase PLSM's specific contributions.
> > - On the IQM results for SPR + PLSM: The overlapping trust intervals between SPR + PLSM, SPR, and DrQ make it difficult to draw definitive conclusions about PLSM's performance improvements. More robust statistical analysis may be needed to support any claims of superiority.
> > - Hyperparameter sensitivity: Supporting Figure F highlights PLSM's sensitivity to the hyperparameter $\beta$ across a wide range (1e-6 to 5.0). This sensitivity could pose challenges for real-world applications where extensive hyperparameter tuning may not be feasible.
> >
> > While I continue to see value in this paper and maintain my recommendation for acceptance, these remaining concerns prevent me from increasing my score. I believe addressing these points could further strengthen the paper's contributions to the field.

---

> > > ### Author Response · Authors · 2024-08-10
> > > **Thanks for the additional comments**
> > >
> > > We thank the reviewer for raising these additional points. We agree that our contribution could be additionally strengthened by addressing these. As such, we ran an additional experiment where we evaluated Dreamer with and without PLSM on walker-walk in Distracting DMC. We first note that Dreamer does considerably worse than RePO, due to the pixel-reconstruction objective. Nevertheless, we see that adding PLSM strengthens the performance of Dreamer on the distracting DMC task. Dreamer attains a mean score of 460 ($ \pm 66$ CI), whereas Dreamer + PLSM attains a mean score of 530 ( $\pm 57$ CI), averaged over 5 seeds after 1 million environment steps.
> > >
> > > We agree with the reviewer that it’s not easy to draw conclusions about performance across all Atari games based on five seeds. However, in several individual games we see that PLSM brings significantly better scores just based on 5 seeds. For instance, in UpNDown, Pong, Krull, Freeway and Assault we observe significantly better human normalized mean scores with just five seeds. We will therefore tone down our claims in the section discussing the Atari results.

---

### Official Review · Reviewer_gEcM · 2024-07-13

**Soundness:** 3
**Presentation:** 2
**Contribution:** 2
**Rating:** 5
**Confidence:** 4

**Summary:**

The paper addresses learning a world model with state-invariant dynamics. To this end, it proposes Parsimonious Latent Space Model (PLSM), which introduces an information bottleneck to the additive dynamics residual. The influence of the state on the dynamics is summarized in the bottleneck variable, whose norm is regularized in the loss function. The proposed PLSM is combined with contrastive learning (Contrastive World Model) to demonstrate its advantage in accuracy, generalization, and robustness for future state prediction. The paper also conducts experiments on PLSM with Self-Predictive Representation and TD-MPC to show the benefits of PLSM to downstream control.

**Strengths:**

- The idea of learning state-invariant latent dynamics, and the way of introducing an information bottleneck to achieve it, are novel. This work contains an interesting exploration toward this idea.
- Extensive experiments are done to support the proposed methods, including two settings in various environments.

**Weaknesses:**

- There seem to be important flaws in the proposed PLSM.
  - First, the math derivation in Appendix A contains several loose ends. Why replacing $p(h|a)$ by $q(h|a)$ in (13) results in a upper bound? Also, why is (13) an equality? Why is it reasonable to assume $q(h_t|a_t)$ is the standard Gaussian distribution and $h_t$ is another isotropic Gaussian? Why does $(15)$, the $KL_D[p(h_t |z_t , a_t ) || q_{\theta} (h_t |a_t )]$ have no dependence on $z_t$ at all? And even if we accept the Gaussian assumption, why immediately after (15), does $h_t$ become a deterministic quantity in Line 440?
  - Second, regardless of the math derivations, I have concerns about the fundamental validity of the regularization on $||h_t||$. It just seems to me that the norm of $h_t$ can be made arbitrarily small without changing the dynamics. For example, let $h_t = 0.0001 f_{\theta}(z_t, a_t)$ and let $\Delta_t = d_{\theta}(10000 h_t, a_t)$. Then, $||h_t||$ becomes 10000 times smaller, but the dynamics remains the same. The constants 0.0001 and 10000 can be absorbed in the parameter $\theta$. This flaw undermines my confidence in the proposed approach.
- The experiments do not appear to show that the proposed PLSM has a significant advantage, especially in the control setting. Actually, Figure 7 shows with PLSM, the performance drops in more environments than the performance improves.
- The writing has much room for improvement.
  - First, the experiments is organized in three sections, namely Sections 3-5. It is unclear to me why Section 3 is a standalone section (and if so, is it better to call it "Experiment setup", rather than "Experiments"). I could be wrong, but it actually seems to me that Section 4 is in parallel to Section 4.1, both of which may be better organized as subsections of Section 3.
  - Some figures are dangling in the paper without being referenced, e.g., Figure 1.
  - Equations are better referenced in parentheses like (1)(2); in LaTeX the command is `\eqref`.
  - Some other minor problems are detailed in the "Questions" section below.

**Questions:**

- Line 9: "minimizing the dependence the state has on dynamics". Logically, maybe "the dependence the dynamics has on the state", or "the impact the state has on the dynamics"? E.g., in Line 34, it also says "the degree to which the predicted dynamics depend on $z_t$".
- In Equation (5), should there be a comma between $z_t$ and $a_t$?
- Line 111: Could the authors elaborate a bit on how (9) is paired with (8)?
- In Figure 5, in the x labels, it is suggested that the math symbols are added for better clarity, e.g., "Latent state $z_t$".
- Line 185: "We evaluated the PLSM's effect": remove "the" to be consistent with other places in the paper?

**Limitations:**

Limitations are commented above. No societal impact needs to be particularly addressed.

---

> ### Author Rebuttal · Authors · 2024-08-06
>
> We thank the reviewer for their thorough and detailed review of our paper. Based on the reviewer’s suggestions, we have clarified the details of the regularization in the appendix, and performed several analyses to show that our regularization works in a principled way. In short, we find that our regularization does not increase weight norms in order to compensate for our sparsity term (see Supporting Figure D). We also show that PLSM improves considerably upon existing methods in visual control tasks with distracting backgrounds (see supporting Figure A), suggesting that our model offers more gains in environments where the agent needs to learn to ignore irrelevant cues. We would like to thank the reviewer again for raising these great points - we believe that addressing them has improved our paper considerably. We answer the reviewer’s questions in more detail below.
>
> >First, the math derivation in Appendix A contains several loose ends
>
> We thank the reviewer for making us aware of a mistake in equation 13 which should say that the mutual information is upper bounded (instead of equivalent to) the variational approximation. For the variational distribution we can choose the variational family ourselves, and we choose the Gaussian distribution. In equation 14, the KL divergence does depend on z, but the prior of h is invariant to z. Lastly, instead of making our model stochastic, we implement a deterministic variant following [1]. Here, L2 regularization is also applied to the dynamics weights, which we have added to prevent unbounded optimization (see Supporting Figure E).
>
> >I have concerns about the fundamental validity of the regularization on $||h_t||$
>
> The reviewer raises a good point: the query representation can be made arbitrarily small by increasing the magnitude of the dynamics weights. We offer several analyses to address this point. First, we show that the norm of the dynamics weights of PLSM does not increase substantially across several datasets, whereas the norm of the query representation is often several orders of magnitude smaller (see Supporting Figure D). Secondly, while we do not see this compensation in our experiments, it can be prevented by penalizing the L2 norm of the dynamics weights. We also evaluate top-k sparsification as an alternative sparsification method. These methods perform comparably and sometimes better in the three datasets we tested them on (see Supporting Figure D).
>
> >The experiments do not appear to show that the proposed PLSM has a significant advantage
>
> We compared PLSM against another robust RL algorithm, RePO [2], on five challenging visual control tasks with distracting backgrounds (distracting DMC [3]). In this setting, where the agent needs to learn to ignore distracting cues, we see considerable improvements by adding our regularization to RePO (see Supporting Figure A, scores are averaged across 5 seeds, with shaded regions representing standard errors). This makes sense, as PLSM compresses away aspects of the environment whose dynamics are unpredictable given the agent’s actions. We further optimized the regularization parameter $\beta$ in Atari, and find that $\beta = 5$ works the best, improving performance in Atari from 61.5 % human normalized score for SPR, to 67% with PLSM (see Supporting Figure C).
>
> > It is unclear to me why Section 3 is a standalone section
>
> We thank the reviewer for raising this point. We have decided to follow the reviewer’s advice and combine section 3 and 4 into one section, as they belong together. We leave the RL experiments in their own section.
>
> >Some figures are dangling in the paper without being referenced
>
> Thank you for pointing this out, we now reference the first figure in the following sentence in the introduction.
>
> **"Here we explore the possibility of compressing states and dynamics jointly to learn systematic effects of actions (Fig 1.)"**
>
> >Line 111: Could the authors elaborate a bit on how (9) is paired with (8)?
>
> To clarify the workings of our model, we expounded on the connection between (8) and (9) in the paper.
>
> **"To apply our regularization on the contrastive dynamics model, we simply add the norm of the query representation to the contrastive loss, similarly to equation (8)."**
>
> We also make minor adjustments to the text as per the reviewer’s recommendation, like referring to PLSM in a consistent way throughout the paper, refer to equations using \eqref, using math symbols in Fig. 5, add a comma in equation (5), and change our formulation in line 9 to “the impact the state has on the dynamics”.
>
> References:
>
> [1] From Variational to Deterministic Autoencoders, Ghosh et al., ICLR 2020
>
> [2] RePo: Resilient Model-Based Reinforcement Learning by Regularizing Posterior Predictability, Zhu et al., NeurIPS 2023
>
> [3] The Distracting Control Suite – A Challenging Benchmark for Reinforcement Learning from Pixels, Stone et al., ArXiv, 2021

---

> > ### Comment · Area_Chair_ihF4 · 2024-08-13
> > **Please address the author's rebuttal**
> >
> > Dear gEcM, please try to respond to the author's rebuttal and elaborate if it made you reassess your opinion.
> >
> > Thank you,

---

> > ### Comment · Reviewer_gEcM · 2024-08-14
> > **Thanks for the rebuttal**
> >
> > I appreciate the authors' response, which indeed addresses several of my concerns, especially my biggest concern about the validity of regularizing $\|h_t\|$ without considering $d_{\theta}$. I do feel the authors need to discuss this point and their solution in the paper. I am also more confident about the mathematical derivation. Earlier on I misunderstood (15) as not depending on $z_t$, but realizing $h_t$ is actually $f_{\theta}(z_t, a_t)$, I now see how (15) indeed depends on $z_t$ and does follow from (14). I suggest the authors remind the reader this point in the paper to avoid confusion. It may be helpful to also clarify that "the deterministic setup" (Line 440) refers to $h_t | z_t, a_t$ is deterministic, and $h_t$ remains to be a random variable and is not deterministic. Overall, I am happy that the authors agree to incorporate many of my suggestions, and I believe an updated version of the paper should be in a much better shape. As a result, I raise my score from 3 to 5.

---

> > > ### Author Response · Authors · 2024-08-14
> > > **Thank you!**
> > >
> > > We thank the reviewer for helping us clarify our approach, and to show the validity of our regularization, we agree that our paper is better as a consequence. We thank the reviewer for raising the score accordingly.

---

### Official Review · Reviewer_PXJB · 2024-07-30

**Soundness:** 3
**Presentation:** 3
**Contribution:** 3
**Rating:** 7
**Confidence:** 3

**Summary:**

The authors propose an approach to learn latent dynamics from high dimensional observations. Their method seeks to minimize the mutual information between the current latent state and the change in the latent state conditioned on the next action, which they argue minimizes the dependence that the state representation has on the dynamics. They evaluate their method on predicting future states, and on model-based and model-free reinforcement learning. They show that their method improves accuracy and downstream task success in difficult continuous control environments, and also helps in some Atari game settings.

**Strengths:**

The methods and experimental setup were clear and well-presented. The authors performed extensive experiments both studying the properties of their learned latent space and on downstream task performance. Their method appears to perform extremely well when the state of the world is fully controllable by the agent.

Although the method does not improve in all settings in which the authors evaluated it, they do not tune parameters for each environment, leaving room for potential improvement.

**Weaknesses:**

The biggest glaring weakness is in the continuous control experiments. The graphs cutoff at 10^5 environment steps which is much shorted than the original TD-MPC paper, and not till convergence for most tasks. This begs the question what the method's asymptotic performance is compared to the original method.

In addition to the continuous control experiments, it would have been nice to see some explanation or hypothesis as to why some of the Atari  games performance degraded so much compared to not using their method.

**Questions:**

Have you run the TD-MPC experiments longer? This is a super important experiment to run and should be included in the paper

**Limitations:**

Yes

---

> ### Author Rebuttal · Authors · 2024-08-06
>
> We thank the reviewer for the thoughtful review, and pointing out ways in which we can improve the paper. To address the reviewer’s concern about the low number of training steps, we extended the training runs for the DMC tasks where performance had not converged, and see that PLSM retains its advantage (see Supporting Figure B). We also tuned the regularization strength for Atari and find that $\beta = 5$ yields the best performance overall, increasing mean normalized scores from 61.5 % for SPR, to 67% with PLSM (see Supporting Figure C). We also offer an explanation for why PLSM yields worse performance in some Atari games, which we have added to the Appendix. We thank the reviewer again for their feedback, as we think addressing these criticisms has helped us clarify the contribution of our model, and improved our paper. We answer the reviewer’s questions in more detail below:
>
> >Have you run the TD-MPC experiments longer?
>
> We extended training runs by another 500k steps for acrobot-swingup and finger-turn hard. We extended the runs for humanoid-walk by another 250k steps since the algorithms were already close to convergence. Results are shown in Support PDF Figure B, averaged over 5 seeds, with shaded regions representing standard error of the mean. We see that PLSM retains the advantage, with higher converged performance for acrobot-swingup, and faster convergence for finger-turn hard. After an additional 250k steps, PLSM is still superior to TDMPC in humanoid-walk.
>
> We sought to strengthen the continuous control experiments further by testing PLSM in the Distracting DeepMind Control Suite environments [1], where the static background in DMC is replaced with a video that is irrelevant for solving the task. We built upon RePO [2] as a baseline, using the official implementation. RePO uses a GRU for latent dynamics prediction, and we incorporated our regularization into it by learning a compressed query representation $h_t$, and passed this alongside the action to the GRU, making it softly state-invariant. Using PLSM, we see important performance gains in several tasks, suggesting that our regularization can be beneficial in environments where the agent needs to learn to ignore distracting cues (see Supporting Figure A).
>
> >It would have been nice to see some explanation or hypothesis as to why some of the Atari games performance degraded
>
> We ran PLSM on a subset of the Atari games with varying degrees of regularization strengths to see if over regularization could explain why PLSM degrades performance in some games. See Support PDF Figure F, error bars representing standard error of the mean. We see that in games where important aspects of the environment are not controllable by the agent, weaker regularization is beneficial. For instance, in Boxing, the movement of the opponent is hard to predict, and here having a lower regularization strength is advantageous. We have added the following paragraph in the appendix:
>
> **"We see that in games where important features of the environment are not controllable by the agent, weaker regularization is beneficial. For instance, in Boxing, the movement of the opponent is hard to predict, and here having a lower regularization strength is advantageous"**
>
> References:
>
> [1] The Distracting Control Suite – A Challenging Benchmark for Reinforcement Learning from Pixels, Stone et al., ArXiv, 2021
>
> [2] RePo: Resilient Model-Based Reinforcement Learning by Regularizing Posterior Predictability, Zhu et al., NeurIPS 2023

---

> > ### Comment · Reviewer_PXJB · 2024-08-12
> > **Rebuttal Response**
> >
> > I would like to thank the authors for taking the time to respond to my concerns and answer my questions. Given the additional experimental evaluations I am increasing my score to accept.

---

> > > ### Author Response · Authors · 2024-08-13
> > > **Thank you**
> > >
> > > We thank the reviewer for their helpful review, which we are sure has allowed us to make important improvements to it, and are grateful for the increase in score.

---

### Author Rebuttal · Authors · 2024-08-06

We would like to thank all of the reviewers for the time and effort put into providing thoughtful and constructive feedback on our paper. Reviewers generally found our method novel and interesting.

* Reviewer PXJB wrote that ‘The authors performed extensive experiments’ and that our method ‘appears to perform extremely well when the state of the world is fully controllable by the agent.’
* Reviewer gEcM wrote that ‘idea of learning state-invariant latent dynamics [...] are novel’ and that ‘Extensive experiments are done to support the proposed methods’

* Reviewer itzz said that our experiments ‘cover a rather broad range of problems which showcase the general applicability of the method.’

* Reviewer hbwZ wrote that our method is ‘regularizing the behavior of the latent space dynamic model and allowing for strong generalization.’

* Reviewer VZsj wrote that our paper ‘is well written’, including experimental evaluation ‘done across three very different settings’

At the same time, reviewers raised some important concerns. Reviewers hbwZ, gEcM and VZsj pointed out that PLSM's performance improvements are sometimes modest. Reviewers VZsj and gEcM also wondered if the sparsity of the query representation could be absorbed by having larger weights in the ensuing linear layer. Reviewers also suggested that we evaluate our regularization relative to other robust RL baselines, specifically RePo, ablate the query representations, run DMC experiments for more time steps, and analyze the model’s sensitivity to the regularization strength. We performed a series of experiments to address these points, and show the results in the attached supporting PDF document.

* First, as requested by reviewers itzz, hbwZ, gEcM and VZsj, we show that PLSM can offer even stronger performance improvements in challenging visual control tasks with distracting visual cues (see Supporting Figure A).  We compare PLSM to RePO [1], a robust model-based RL algorithm, on tasks from the Distracting DeepMind Control Suite [2], where the background is replaced with a distracting, irrelevant video. We incorporated our regularization on top of RePO using the official implementation. Evaluating the models across five environments with five seeds each, we see that using our regularization offers considerable improvements in scores. This makes sense, since PLSM compresses away aspects of the environment that are hard to predict given the agent’s actions.

* We show empirically that PLSM does not compensate for the sparsity by adopting larger weights (requested by VZsj and gEcM, see Supporting Figure D). We also propose two alternative implementations of PLSM where artificially increasing the norm of the weights cannot compensate for our regularization. Both of these alternative implementations perform equally well, and sometimes better than the original PLSM (see Supporting Figure E).

* We extend the number of training steps in the DeepMind Control Suite experiments, showing faster and improved convergence (requested by PXJB, see Supporting Figure B).

* We provide more detailed results in Atari, along with Interquartile Mean estimates (requested by itzz, see Supporting Figure. C).

* We analyze how the regularization strength impacts performance in Atari, and offer an explanation as to why performance drops for some games (requested by itzz and PXJB, see Supporting Figure F).

* We show that removing the query net substantially decreases PLSM performance across several datasets (requested by VZsj, see supporting Figure E).

* We have cleaned up notation and clarified how we arrive at our regularization (requested by gEcM).



All of these analyses and results are outlined in detail in our responses to the individual reviews below, where we also address more specific questions reviewers had. We again want to thank the reviewers for their time and for actively engaging in the review process.

[1] RePo: Resilient Model-Based Reinforcement Learning by Regularizing Posterior Predictability, Zhu et al., NeurIPS 2023

[2] The Distracting Control Suite – A Challenging Benchmark for Reinforcement Learning from Pixels, Stone et al., ArXiv, 2021

---

> ### Author Response · Authors · 2024-08-14
> **Summary of discussion period**
>
> We would like to express our gratitude to all the reviewers and AC for engaging with our work and providing valuable feedback. The reviewer’s comments were indispensable for improving our paper. The reviewers made important suggestions regarding the validity and impact of our regularization method. To address these we first showed that PLSM works in a principled way. We also extended our experimental evaluation to include challenging visual control tasks with distractions and another baseline that benefits from our regularization. **Now all five reviewers recommend accepting our paper, with an average score of 5.8**. In sum, we think implementing the changes and extra evaluations suggested by the reviewers made our paper much stronger, and we are glad that this is reflected in the increased scores. Thank you all again!

---

### Decision · Program_Chairs · 2024-09-25

**Decision:**

Accept (poster)

**Comment:**

Learning latent state representation is crucial for scaling reinforcement learning to real-world problems as these often operate in this limit. In this work, the authors proposed a new and practical approach for learning latent dynamics in high-dimensional spaces. The new method, PLSM, learns the latent state representation by learning the action conditional mutual information between the state  and residual changes in the representation.  The authors conducted thorough empirical investigations and managed to show their method improves performance of model free and model based algorithms in standard reinforcement learning benchmarks.

In summary, this work advances our understanding of latent state representation learning and explores a novel algorithmic idea. Reviewers and I agree that it is valuable to the community, and I support accepting it.